

# Northern Hemisphere Stratospheric Polar Vortex Morphology under Localized Gravity Wave Forcing: A Shape-Based Classification

Sina Mehrdad[1,2], Sajedeh Marjani[1], Dörthe Handorf[3], and Christoph Jacobi[1]

[1]Leipzig Institute for Meteorology, Leipzig University, Stephanstr. 3, 04103 Leipzig, Germany
[2]Department of Earth Science and Engineering, Imperial College London, London SW7 2AZ, UK
[3]Alfred Wegener Institute for Polar and Marine Research, Research Unit Potsdam, D-14473Potsdam, Telegrafenberg A43, Germany

**Correspondence:** Sina Mehrdad (s.mehrdad@imperial.ac.uk)

**Abstract.** The Northern Hemisphere stratospheric polar vortex (SPV) response to localized gravity wave (GW) forcing remains poorly understood, particularly in terms of its detailed morphology. Here, we investigated geometry-specific impacts of enhanced orographic GW drag in three hotspot regions, the Himalayas, Northwest America, and East Asia, using ensemble simulations with the high-top UA-ICON global circulation model. By classifying daily SPV geometries into ten distinct clus-

5 ters with a novel unsupervised, shape-based hierarchical clustering framework, we isolated geometry-specific responses using the class contribution method. Our results showed that all hotspot forcings consistently reduce planetary wave 1 (PW1) amplitude and induce a PW1-like displacement of the SPV core, though spatial patterns vary with hotspot location. This response manifested as negative geopotential height (GPH) anomalies within the forced region and positive anomalies to the north, indicating localized SPV edge mixing. The response was also sensitive to the forcing's latitudinal position: the Himalayas, as

the southernmost hotspot, produced a deepened vortex, while the more poleward Northwest America and East Asia forcings showed similar patterns with greater intrusion of positive GPH anomalies into the vortex core. The forcing reduced PW1 amplitude both by shifting the frequency of specific clusters and altering the mean structure of the most frequent classes. Our results demonstrate that shape-based clustering combined with the class contribution framework can reveal robust, spatially coherent signals that might otherwise be masked by internal variability, providing a new perspective for understanding SPV

variability and its predictability.

## 1 Introduction

The stratospheric polar vortex in the Northern Hemisphere (SPV from now on) is a dominant cyclonic feature in the stratosphere that surrounds the winter pole, characterized by strong westerlies forming the polar night jet (Waugh et al., 2017). Compared to its Southern Hemisphere counterpart, the SPV exhibits greater variability due to stronger wave activity in the Northern

Hemisphere, primarily driven by the extensive land-ocean contrasts and prominent orographic features (Waugh and Randel, 1999). The variability of the SPV can substantially influence tropospheric weather and climate through downward coupling





processes (Haynes et al., 1991; Baldwin and Dunkerton, 2001; Hardiman and Haynes, 2008; Hitchcock and Simpson, 2014; Kidston et al., 2015; Scaife et al., 2021).

Large-scale planetary waves (PWs) are one of the major drivers of the stratospheric dynamics (Andrews and Mcintyre, 1976).
These waves significantly influence the state and morphology of the SPV (Waugh, 1997; Karpetchko et al., 2005; Günther et al., 2008; Albers and Birner, 2014). PWs typically have large horizontal scales—most commonly zonal wavenumber 1 (PW1) or 2 (PW2)—and can propagate vertically into the stratosphere, where they distort and decelerate the vortex (Riese et al., 2002). In essence, the stratospheric vortex's general shape is a direct imprint of these waves. PW1 induces a single ridge–trough pattern, displacing the vortex off the pole, while PW2 generates two ridges and two troughs, elongating the vortex into an elliptical structure or even splitting it into two distinct lobes (Finke et al., 2025). The breaking of PWs in the stratosphere is broadly recognized as the main driver of sudden stratospheric warmings (SSWs), abrupt and significant rises in polar stratospheric temperatures during winter, associated with a decrease or even reversal of the typical zonal mean westerly winds (Baldwin et al., 2021).

In addition to PWs, gravity waves (GWs) also play a crucial role in shaping wintertime stratospheric dynamics and the structure of the SPV by transferring momentum and energy upward from the troposphere to middle atmosphere regions where they dissipate (Andrews et al., 1987; Fritts and Alexander, 2003; Yiğit and Medvedev, 2016). GWs can originate from various sources, broadly categorized as orographic, generated by airflow over major mountain ranges (Smith, 1980; Nastrom and Fritts, 1992), and non-orographic, arising from processes such as deep convection, jet streams, frontal systems, and shear instabilities (Eckermann and Vincent, 1993; Alexander et al., 1995; Bühler et al., 1999; Plougonven and Zhang, 2014). GW activity exhibits a pronounced zonal asymmetry, with well-defined hotspot regions where GW amplitudes and associated momentum drag are particularly strong (Ern et al., 2004; Fröhlich et al., 2007; Hoffmann et al., 2013; Schmidt et al., 2016). These hotspots are typically episodic and intermittent in nature and are commonly linked to prominent orographic regions such as the Himalayas, the North American Rockies, and East Asia (Hertzog et al., 2012; Hoffmann et al., 2013; Wright et al., 2013; Šácha et al., 2015; Kuchar et al., 2020; Gupta et al., 2024; Hozumi et al., 2024).

The location and distribution of GW drag exert a strong influence on the state of the SPV (Samtleben et al., 2019, 2020a, b; Kuchar et al., 2020; Sacha et al., 2021; Mehrdad et al., 2025a). GWs can affect the SPV by modulating PW behavior in the stratosphere through mechanisms collectively referred to as compensation mechanisms (Cohen et al., 2013, 2014; Sigmond and Shepherd, 2014; Karami et al., 2022). These include the ability of GW drag to disrupt the vortex edge and locally generate PW activity at breaking regions, thereby reshaping the SPV (Coy et al., 2024). The extent of GW influence on stratospheric dynamics is highly sensitive to the spatial and temporal patterns of the resulting GW drag (Boos and Shaw, 2013; Shaw and Boos, 2012; Šácha et al., 2016; Kuchar et al., 2020), with a particularly strong dependence on how the GW drag aligns in phase with PWs (Samtleben et al., 2019, 2020a; Mehrdad et al., 2025a).

GW activity contributes substantially to differences seen in inter-model comparisons of stratospheric dynamics (Sigmond et al., 2023; Kuchar et al., 2024). The morphology of the SPV plays a crucial role in shaping the distribution of GW activity by modifying the background wind and temperature fields in the stratosphere (Andrews et al., 1987). In turn, GW activity can feed back on the SPV, altering its shape and dynamical behavior (Coy et al., 2024; Kuchar et al., 2024). Despite growing evidence





that regional GW hotspots modulate the middle atmospheric dynamics (Šácha et al., 2016; Samtleben et al., 2019, 2020a; Kuchar et al., 2020; Sacha et al., 2021), their geometry-specific and climate-scale impacts on the SPV remain poorly quantified. They are particularly important because the zonally asymmetric nature of GW forcing means that its dynamical effects on the

SPV can vary significantly depending on the underlying SPV geometry and state. Previous studies have documented hotspot-induced changes in the zonal-mean context (Samtleben et al., 2020a; Kuchar et al., 2020; Sacha et al., 2021), but none have systematically examined these changes in the context of SPV morphology. Such a perspective is critical, as the SPV's structure governs downward coupling to the troposphere and influences the likelihood of extreme events such as SSWs (Baldwin and Dunkerton, 2001; Charlton and Polvani, 2007; Seviour et al., 2013; Messori et al., 2022; Ding et al., 2023).

Here we build on the high-top UA-ICON experiments of Mehrdad et al. (2025a) but extend the analysis in three pivotal ways. (i) We implement a hierarchical unsupervised, shape-based classification that categorizes the daily SPV state into ten distinct geometry-based clusters. (ii) Leveraging the class-contribution framework introduced by Mehrdad et al. (2024), along with ensemble-based analysis, we isolate the characteristic patterns through which enhanced GW hotspot forcing modifies the SPV, distinguishing them from internal variability. (iii) We investigate how these geometry-conditioned responses feed back onto

broader stratospheric dynamics. By focusing on SPV morphology, this approach extends the zonal-mean analysis of Mehrdad et al. (2025a), offering a new lens through which to understand the influence of non-zonal GW forcing on Arctic stratospheric variability.

## 2    Data and methodology

This section outlines the simulation setup, datasets, and analysis procedures used in this study.

### 2.1    Model simulations

We employ the same set of ensemble climate simulations as described in Mehrdad et al. (2025a), using the upper-atmosphere configuration of the ICON general circulation model (UA-ICON; Borchert et al. 2019). UA-ICON is a high-top model that extends well above the stratopause and features fine vertical resolution, meeting the requirements for realistically simulating SPV behavior (Wu and Reichler, 2020; Zhao et al., 2022; Kuchar et al., 2024). The model has been shown to realistically

represent stratospheric dynamics and key aspects of wave-driven troposphere–stratosphere coupling (Köhler et al., 2023; Kunze et al., 2025; Mehrdad et al., 2025a). The simulations were performed with the R2B4 grid (approximately 160 km horizontal resolution), 120 vertical levels extending up to  150 km, and were driven by annually repeating present-day climatological boundary conditions.

To investigate the impact of regionally intensified orographic GW forcing, we conducted a control simulation (C) and three

sensitivity experiments. The control run used the default model configuration. In the sensitivity experiments, the GW drag produced by the subgrid-scale orographic (SSO) scheme (Lott and Miller, 1997) was enhanced by a factor of 10 in three GW hotspot regions—the Himalayas (HI), Northwest America (NA), and East Asia (EA)—while the low-level component of the SSO drag, associated with flow blocking and wake effects, remained unchanged. The forced regions are defined as follows: EA





spans $30° - 60°N$ and $110° - 175°E$; NA spans $30° - 60°N$ and $100° - 130°W$; and HI spans $25° - 45°N$ and $70° - 100°E$.
Each experiment, including the control run, comprises a six-member ensemble of 30-year simulations under identical boundary conditions. For full details on the model configuration, ensemble design, and GW forcing strategy, see Mehrdad et al. (2025a).

## 2.2 Data

In this study, we analyzed daily model output for the extended boreal winter season (November through March, NDJFM), focusing exclusively on the Northern Hemisphere, considering data poleward of $0°$ latitude. Daily values were computed as the
95 average across all model time steps within each simulation day. The first year of each simulation was considered as the model spin-up time and was therefore excluded from the analysis.

To characterize the stratospheric polar vortex, we used the geopotential height (GPH) field at the 10 hPa level. These GPH fields were interpolated onto a Lambert azimuthal equal-area projection centered on the North Pole (Snyder, 1987), using a horizontal grid spacing of $1.5°$, which closely matches the native resolution of the model. We also used temperature and
100 horizontal wind fields to derive zonal-mean quantities and Eliassen–Palm (EP) flux and its divergence (Andrews et al., 1987).

## 2.3 Classification of SPV geometry

We used GPH at the 10 hPa level to represent the SPV. The morphology of the SPV can significantly influence, and be influenced by, the regional interactions of GWs with the atmospheric flow. To explore these interactions, we developed an algorithm to classify the SPV geometry in an unsupervised manner and analyzed the impact of regional GW hotspots across
different vortex geometry categories. The classification process involves several steps. First, we extracted the vortex boundaries and derived a compressed feature space to represent the SPV geometry. Then, we classified the vortex geometry based on this feature space. In the following sections, we provide a detailed explanation of each step in this process.

### 2.3.1 SPV boundary detection and feature extraction

The vortex boundaries were extracted from the daily 10 hPa GPH fields of all simulations during the extended winter period
(see an example in Figure 1). We retained only data northward of $25°N$. For each daily field, we identified the lowest 18th percentile of GPH values as the initial threshold for delineating the SPV boundaries, shown by the white line in the left panel of Figure 1. The 18% threshold was chosen empirically to capture a suitable range of vortex geometry.

Next, we defined each closed boundary in the daily field with more than 10 boundary pixels as an SPV boundary object. A daily field could therefore contain zero, one, or multiple boundary objects. For each SPV boundary object identified in a daily
field, we extracted Fourier descriptors (Zahn and Roskies, 1972; Persoon and Fu, 1977) as follows:

1. Forming the complex signal:

   We treated the two-dimensional boundary coordinates $(x_k, y_k)$ in the projected system as a complex signal,

$$z_k = x_k + i y_k, \tag{1}$$





where $i$ is the imaginary unit and $k = 1, \ldots, N$ indexes the boundary pixels along the closed contour.

2. Applying the FFT:

We performed a Fast Fourier Transform (FFT) (Cooley and Tukey, 1965) on this complex signal to obtain its Fourier coefficients. We then discarded the DC component (the zero-frequency term) and retained only the 10 largest remaining coefficients. All other coefficients were set to zero. The previously introduced limit of 10 boundary pixels ensures that each object is sufficiently large to be meaningfully represented by 10 harmonics.

3. Storing Fourier Descriptors:

Because each of the 10 retained coefficients is complex, we stored the respective real and imaginary parts as 20 real values as the Fourier descriptors (Zahn and Roskies, 1972; Persoon and Fu, 1977).

In addition to the 20 Fourier descriptors, we recorded:

– Boundary size: the total number of boundary pixels.

– Mass center: the center of the boundary object, reported as $(x_{\mathrm{center}}, y_{\mathrm{center}})$.

– Time feature: the relative day of the season with respect to January 1 (e.g., November 1 is $-61$, January 10 is $+9$, etc.).

For each SPV boundary object, we therefore derived a feature vector comprising a total of 24 features. The first two features represent the coordinates of the vortex center, the third feature encodes time, features 4 to 23 consist of 20 Fourier descriptors, and the final feature represents the boundary size. Since the focus of this analysis is the SPV shape, we did not include a feature 135 directly representing the strength or depth of the vortex.

This feature space provides a compressed yet rich representation of the SPV state. The Fourier descriptors primarily capture the overall geometry of the vortex boundary. Lower-order harmonics encode global shape attributes, such as circularity versus elongation, while their phases indicate boundary orientation. The boundary size reflects the spatial extent of the vortex, the center encodes its location in a polar projection, and the time feature captures seasonal progression.

Moreover, by performing an inverse FFT with only the retained 10 largest coefficients (excluding the DC component) and translating the result according to the boundary's mass center, we can reconstruct a smoothed representation of the original boundary (see the red line in the right panel of Figure 1). This reconstruction illustrates how the features effectively capture the main structure of the SPV boundary object.

Note that, before performing any clustering or classification, we do not normalize the Fourier descriptors to preserve infor-145 mation about both scale and geometry. However, we rescale the $(x_{\mathrm{center}}, y_{\mathrm{center}})$ coordinates of the boundary center to match the maximum variance among all features in the dataset. Specifically, we calculate the variance of each individual feature across the whole dataset and identify the maximum variance. The central coordinate features are then normalized to have the same variance as this maximum. This adjustment ensures that the location of the boundary center has a comparable influence on the feature space as we employed Euclidean distance-based unsupervised classification. By doing so, we emphasize the role 150 of boundary location in the clustering process.



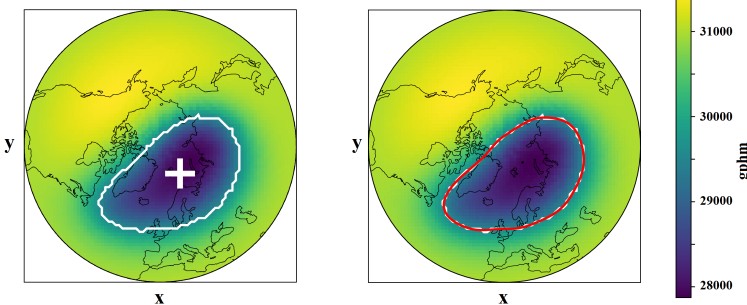

**Figure 1.** Geopotential height (GPH) fields at 10 hPa in GPH meters for a single day (19 February of the second year after the simulation start, corresponding to time feature +49) from the control simulation ensemble 1, shown in a Lambert azimuthal equal-area projection. The x and y axes represent the projection coordinates. The vortex boundary, defined as the lowest 18th percentile of the field values, is outlined with a white line in both panels, while the vortex center of mass $(x_{\mathrm{center}}, y_{\mathrm{center}})$ is indicated by the white cross in the left panel. The right panel also shows the reconstructed vortex boundary, derived from the extracted features, represented by the red line.

### 2.3.2 SPV classification

We devised a hierarchical scheme to classify daily SPV states at 10 hPa. First, we categorized the daily data points based on the number of distinct SPV boundary objects in each daily field. Daily fields containing either zero or more than two SPV boundary objects were classified as unstable SPV and assigned to a global cluster labeled C1. This initial partitioning left two remaining categories for further analysis: single-SPV (exactly one boundary object) and split-SPV (exactly two boundary objects). Within each of these categories, we then applied hierarchical clustering using Ward's method (Ward Jr, 1963; Johnson, 1967), which iteratively merges clusters to minimize the increase in within-cluster variance.

In the single-SPV category, each daily field is represented by a 24-dimensional feature vector. We computed the Euclidean distance in this 24-dimensional space to construct the pairwise distance matrix, a prerequisite for Ward's linkage criterion. The resulting dendrogram, shown in the upper panel of Figure 2, is truncated to highlight the last 50 merges. A visual inspection revealed a distinct gap at approximately $12.5 \times 10^4$ on Ward's linkage axis. Cutting the dendrogram at this level yielded five well-separated clusters, labeled C2–C6.

For the split-SPV category, each daily field contains exactly two SPV boundary objects, each represented by a 24-dimensional feature vector. Before computing distances between any two daily fields, we match the boundary objects by identifying the closest pair of mass-center coordinates $(x_{\mathrm{center}}, y_{\mathrm{center}})$. Then, we compute the Euclidean distance between these matched objects in the 24-dimensional feature space, repeat this for the second pair of objects, and finally take the mean of the two object-wise distances as the distance between the two daily fields. We apply Ward's hierarchical clustering to these pairwise distances, and the resulting dendrogram is shown in the lower panel of Figure 2. A threshold of $6 \times 10^4$ yields four well-separated clusters (C7–C10). Thus, our classification framework partitions the SPV states into one unstable-vortex cluster (C1), five single-SPV clusters (C2–C6), and four split-SPV clusters (C7–C10).




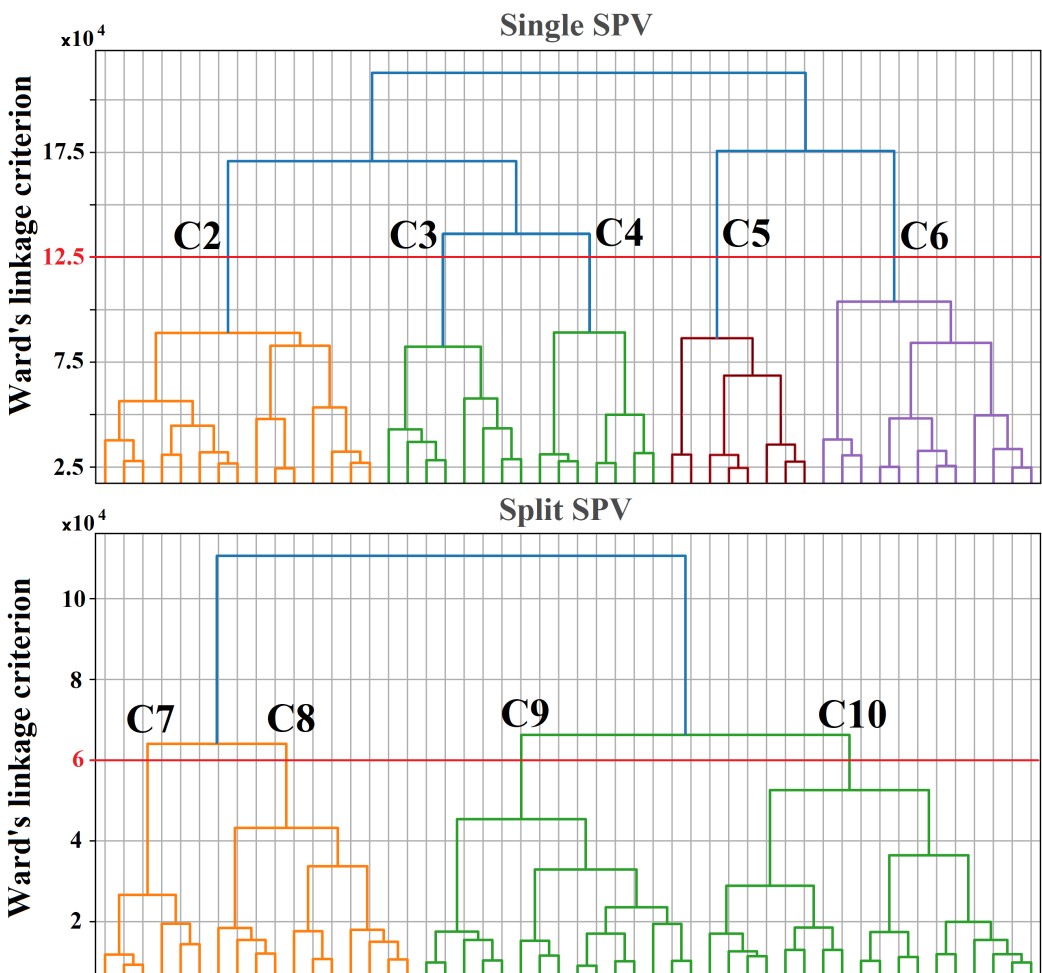

**Figure 2.** Hierarchical clustering dendrograms (Ward Jr, 1963) displaying the last 50 merging steps for the single vortex category (upper panel) and the split vortex category (lower panel). The horizontal axis represents subsets of data points within each cluster branch, while the vertical axis shows Ward's linkage criterion (i.e., the increase in within-cluster variance based on Euclidean pairwise distances in feature space) at which clusters merge. Red horizontal lines in both panels represent thresholds of $12.5 \times 10^4$ (upper panel) and $6 \times 10^4$ (lower panel), which were used to partition the data into five clusters for the single vortex category and four clusters for the split vortex category, respectively. The cluster IDs (C2–C6 and C7–C10) are labeled at the intersection of the branches with the selected thresholds in both panels.

## 2.4 Class contribution

Class contribution quantifies the influence of different clusters—here, clusters of SPV—on anomalies in climatic parameters (Mehrdad et al., 2024). The sum of class contributions across all clusters in a sensitivity simulation yields the total anomaly of





a given parameter. Each cluster's class contribution consists of two main components: Within-Cluster Variability Contribution
(WCVC) and Frequency-Weighted Seasonal Deviation Contribution (FSDC). The WCVC represents how changes in the mean
spatial characteristics of a cluster, such as minor shifts in its geometry or location, contribute to anomalies under the applied
forcing in a sensitivity simulation. In contrast, the FSDC captures how variations in the occurrence frequency of a cluster
influence the overall anomaly. Together, these components help differentiate whether observed anomalies arise from internal
structural changes within a cluster (WCVC) or shifts in its prevalence (FSDC) in response to external forcing. A detailed
methodological formulation is provided in (Mehrdad et al., 2024).

## 3 Results

In this section, we first present the climatology of the control run, followed by the climatological differences in key climate
variables between the control and sensitivity experiments. We then analyze the class centers and their associated mean states,
and describe changes in the occurrence frequency across the different experiments. Finally, we assess the contribution of each
class to the anomalies observed in key climate variables.

### 3.1 Climatology of the control and sensitivity experiments

Figure 3a shows the climatology of GPH at 10 hPa from the control run, along with the reconstructed SPV boundary based on
the mean feature space of the control simulation. The climatological SPV exhibits an elliptical shape, with its center displaced
toward northern Eurasia, consistent with previous findings based on reanalysis data (Kuchar et al., 2024). Panels b–d of Figure 3
illustrate the GPH anomalies at 10 hPa for the HI, NA, and EA sensitivity experiments, respectively, relative to the control
run. The dotted regions indicate where the ensemble mean differences are consistent in sign across at least five out of six
ensemble members, which we refer to as the consistency criterion throughout the manuscript. All sensitivity experiments show
a displacement of the SPV toward North America, although the magnitude of this shift varies among the experiments. The
HI experiment exhibits the most pronounced and consistent shift. A similar pattern is evident in the potential vorticity (PV)
anomalies on the 850 K isentropic surface, as shown in Figure A1 in Appendix A.

Figure 4 presents the climatological structures and anomalies of PW1 (top row) and PW2 (bottom row) based on 10 hPa
GPH fields. Cyan contours in panels (a–c, e–g) represent the climatological PW1 and PW2 patterns from the control run. PW1
is characterized by a high-pressure ridge over the North Pacific and North America and a corresponding low-pressure trough
over Eurasia and the North Atlantic, whereas PW2 shows two ridge centers over the Alaska–Chukchi Sea region and Northern
Europe/Scandinavia (Labitzke, 1981; Andrews et al., 1987).

In all sensitivity experiments, the imposed forcings act against the climatological PW1 pattern (Figure 4; panels a–c), re-
sulting in a reduction of PW1 amplitude across mid- to high latitudes in the Northern Hemisphere (panel d). This amplitude
reduction is strongest in the HI experiment and is consistent across most mid- to high latitudes in both the HI and EA ex-
periments. In contrast, the NA experiment shows consistent amplitude reduction only near the latitudinal edge of the forcing





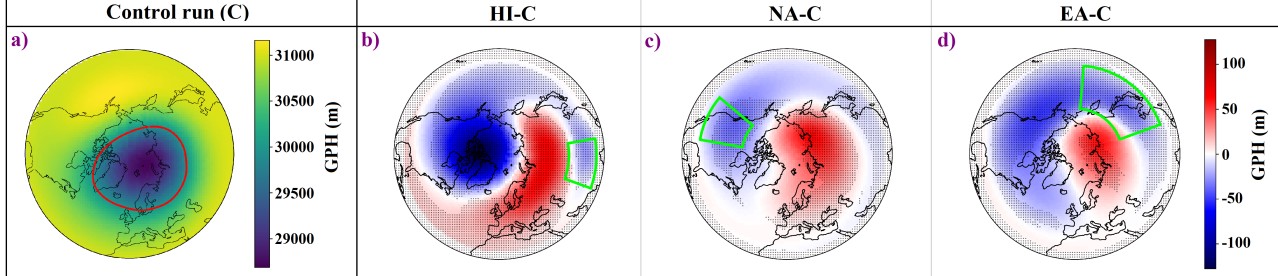

**Figure 3.** (Panel a) Climatology of GPH at 10 hPa from the control run for the extended winter season (NDJFM). The red contour in panel (a) indicates the climatological reconstructed SPV boundary. Panels (b–d) show GPH anomalies for the HI, NA, and EA sensitivity experiments, respectively, relative to the control run, based on ensemble means. Dotted regions in panels (b–d) indicate areas where the anomalies are considered consistent, i.e., at least five out of six ensemble members exhibit the same anomaly sign as the ensemble mean. The green outlines indicate the forced regions in each sensitivity experiment.

region. Additionally, all experiments induce an eastward (positive) phase shift in PW1, which is consistent at high latitudes in the HI and NA experiments, and at lower latitudes in the HI and EA experiments.

For PW2 (Figure 4; panels e–h), the dominant response is again a reduction in amplitude. This reduction is strongest and most spatially consistent in the EA experiment, while in NA it is only partially consistent in the southern portion of the forcing latitudinal range. The HI experiment shows consistent amplitude reductions at low and mid-latitudes, but consistent increases at higher latitudes. Across all experiments, a westward phase shift is observed at lower latitudes (consistent in HI and NA), while an eastward and more consistent shift occurs at higher latitudes.

Figure 5 presents the extended winter climatology of zonal-mean zonal wind and EP flux divergence in the stratosphere. The zonal wind shows a strong westerly jet centered around 60°N near 10 hPa, characteristic of the climatological SPV. The EP flux divergence pattern illustrates the typical wave forcing on the mean flow, with wave drag dominant in the high-latitude stratosphere. Both fields exhibit realistic structures consistent with previous studies and reanalysis-based climatologies (Edmon Jr et al., 1980; Randel et al., 2004).

Figure 6 shows anomalies in the zonal-mean zonal wind (panels a–c) and EP flux divergence (panels d–f) for each sensitivity experiment, relative to the Control run. All sensitivity experiments exhibit a deceleration of the stratospheric westerlies associated with the imposed forcing, extending poleward from the forcing regions (panels a–c). This is accompanied by a general positive EP flux divergence anomaly in the stratosphere (panels d–f), meaning the divergence is reduced relative to the control run, which indicates weaker wave drag (Mehrdad et al., 2025a).

In the HI experiment (Figure 6; panels a and d), westerlies increase consistently throughout the stratosphere at high latitudes, while EP flux divergence anomalies are consistently positive across the midlatitudes and in the high-latitude upper stratosphere, indicating a reduction in wave drag. In contrast, consistent negative EP flux divergence anomalies are observed in the lower to mid stratosphere at high latitudes, implying enhanced wave drag. In the NA experiment (panels b and e), zonal-mean zonal winds exhibit a consistent reduction of the westerlies across the midlatitudes, accompanied by the corresponding



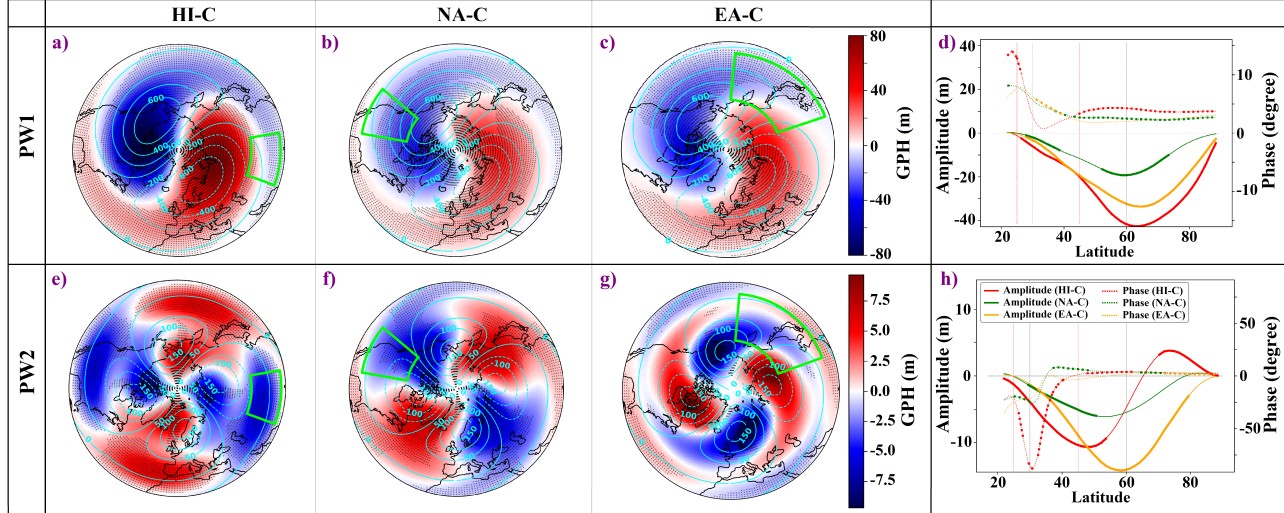

**Figure 4.** Climatology and anomalies of PW1 (top row, panels a–d) and PW2 (bottom row, panels e–h) calculated from 10 hPa GPH fields. Panels (a–c) show PW1 anomalies for the HI, NA, and EA simulations, respectively, with color shading. The corresponding PW1 climatology from the control run is overlaid in cyan contours. Panels (e–g) show PW2 anomalies for the same experiments, also with overlaid control run climatology. Anomalies are based on ensemble means, and dotted regions indicate areas where the anomaly are consistent across ensemble members. Panels (d) and (h) present the latitudinal profiles of PW1 and PW2 amplitude (solid lines, left y-axis) and phase (dotted lines, right y-axis) anomalies, respectively, for each sensitivity simulation relative to the control run. Note the different scaling for PW1 and PW2. Thicker segments indicate latitudes where the anomalies are consistent across ensemble members. Vertical dashed lines (panels (d) and (h)) in red mark the latitudinal extent of the forcing region in the HI experiment, while dashed black lines denote the forcing regions for the NA and EA experiments.

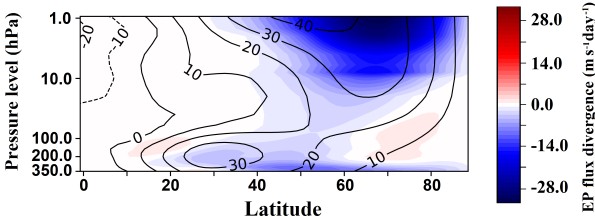

**Figure 5.** Climatological zonal-mean zonal wind (black contours; in m s$^{-1}$) and EP flux divergence (color shading) for the control experiment, averaged over the extended winter season (NDJFM), adapted from Mehrdad et al. (2025a).

consistent positive EP flux divergence anomalies in the same region. In the EA experiment (panels c and f), westerlies increase consistently equatorward of around 40°N but show a consistent decrease north of around 45°N. EP flux divergence anomalies are consistently positive in both mid- and high-latitude regions, with weak but consistent negative anomalies in the lower
stratosphere. We note that in all experiments, the zonal wind decrease related to and poleward of the GW forcing region is





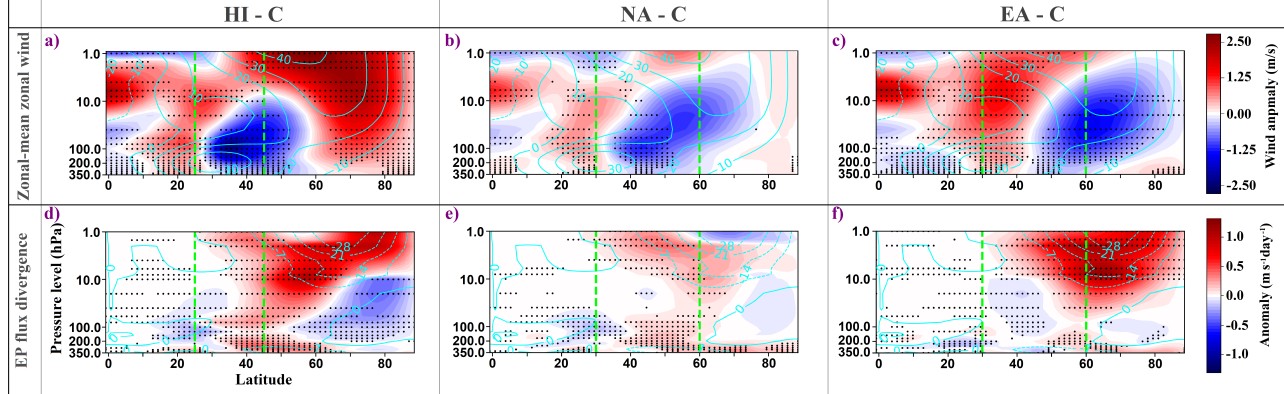

**Figure 6.** Zonal-mean zonal wind (top row; panels a–c) and EP flux divergence (bottom row; panels d–f) anomalies for the HI (left column), NA (middle column), and EA (right column) sensitivity experiments, respectively. Anomalies are computed as the difference between each sensitivity experiment and the control run, based on ensemble means for extended winter (NDJFM). Cyan contours represent the control run climatology of the zonal-mean zonal wind ($m.s^{-1}$; top row) and EP flux divergence ($m.s^{-1}.day^{-1}$; bottom row), with solid lines representing positive values and dashed lines showing negative values. Dotted areas indicate regions where anomalies are consistent across ensemble members. Green dashed lines mark the latitudinal extent of the imposed forcing in each experiment. Adapted from Mehrdad et al. (2025a).

accompanied by a decreasing westward PW forcing, which is outweighed by the GW mean wind forcing (Mehrdad et al., 2025a).

## 3.2 SPV class centers

We used 10 hPa GPH fields to build the SPV shape feature space, which in turn fed the classification described in Section 2.3. Figure 7 displays the resulting GPH class centres, derived from all the control and sensitivity experiments, with red contours overlaid (C2–C10) showing SPV boundaries that were reconstructed from the mean feature vector of each cluster. These reconstructed boundaries closely follow the mean 10 hPa GPH field for their respective clusters, underscoring the physical consistency of the classification.

Class C1 corresponds to an unstable or dissipated vortex, characterized by elevated GPH values near the pole. C2 depicts a centered, deep vortex with slightly elevated GPH over the Pacific sector. C3 shows a moderately displaced vortex shifted toward Eurasia, stretched across the north Eurasia coasts with elevated GPH over North-Western America. C4 represents a weakened, displaced vortex primarily shifted toward the Eurasian continent. The dendrogram in Figure 2 reveals that C2, C3, and C4 are in the same branch, which reflects their similar structural characteristics and the evolution of high-GPH regions from the Pacific in C2 to North America and the central Arctic in C3 and C4 that are closely related in the dendrogram.

C5 shows a deep vortex cluster that stretches from North America to the north of Eurasia. C6 also illustrates a deep vortex cluster that extends across the north Eurasian coasts with a more circular shape. C7 and C8 are split-vortex cases, with the





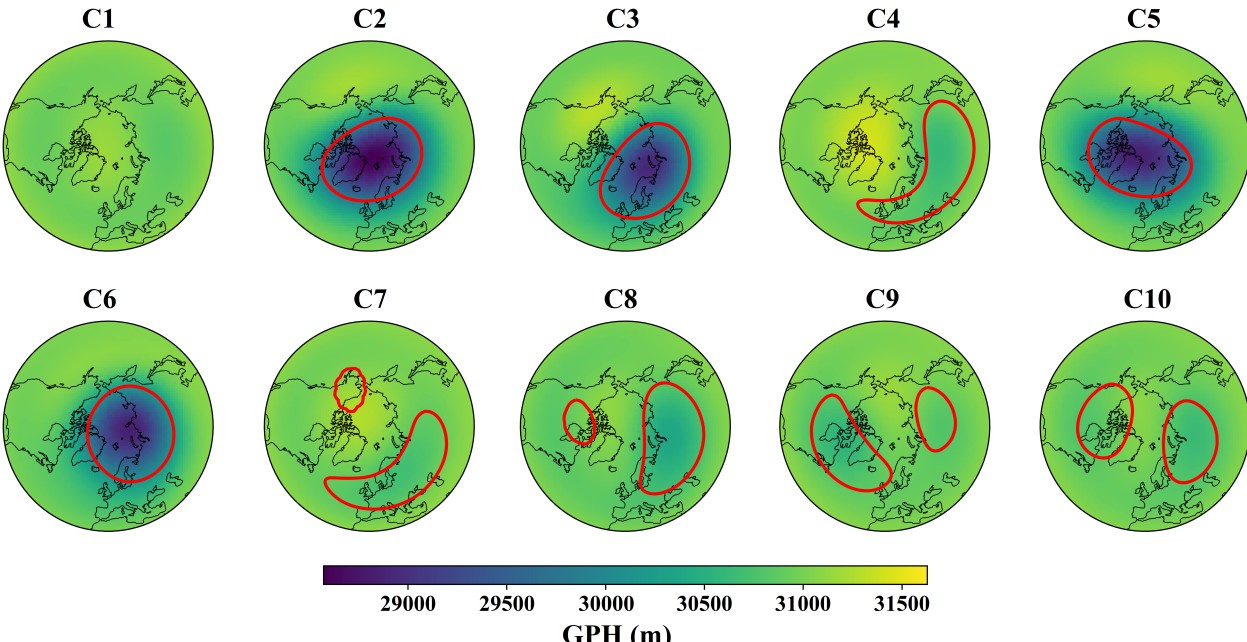

**Figure 7.** Composite 10 hPa GPH fields for SPV classes C1–C10 during the extended winter season (NDJFM), obtained by averaging all daily GPH fields assigned to each cluster across all experiments. Red contours in clusters C2–C10 represent reconstructed vortex boundaries derived from the average feature vectors of all SPV boundary objects associated with each cluster. Cluster C1, which includes unstable or undefined SPV configurations, does not include a reconstructed boundary.

dominant lobe over Eurasia and a smaller one over North America; these two are also closely linked in the dendrogram. In C9, the North American lobe is dominant and extends over the Atlantic, while in C10, both lobes are of comparable size. These latter two classes also form a distinct branch in the dendrogram. Overall, the correspondence between the class centres and the
hierarchical dendrogram confirms the robustness of the shape-based SPV classification.

Figure 8 shows the climatological latitudinal profiles of PW1 and PW2 amplitudes in the control run, along with the corresponding mean amplitudes for each SPV class, calculated across all days labeled to each cluster from all experiments. Cluster C1, representing an unstable vortex, exhibits the lowest amplitudes in both PW1 and PW2. In general, single-vortex classes (C2–C6), with the exception of C4, show higher PW1 amplitudes than the other clusters. Among these, C3 has the strongest
PW1 amplitude, while C5 has the weakest one. This is consistent with the vortex morphology. C3 has the most displaced vortex center, whereas C5 shows a more centered structure.

For PW2 (Figure 8 right panel), C9 has the highest amplitude overall, reflecting its clear two-vortex structure. Among the single-vortex clusters (C2–C6, excluding C4), C5 exhibits the highest PW2 amplitude, likely associated with its elongated vortex shape. In contrast, C6 and C3 show the lowest PW2 amplitudes, which aligns with their relatively circular vortex





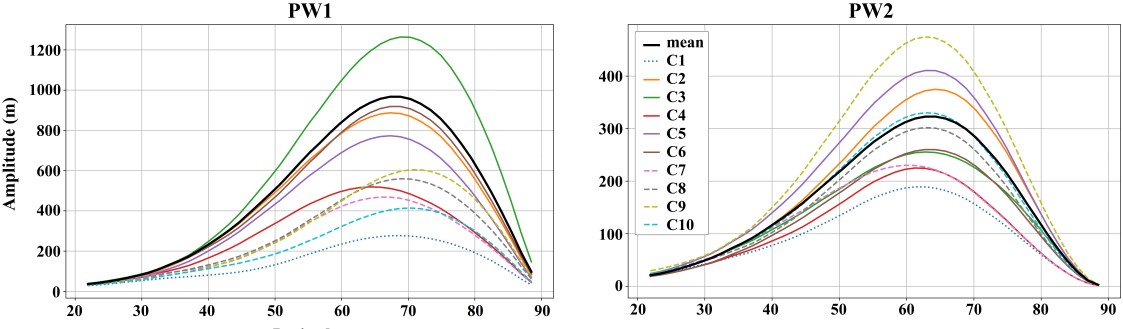

**Figure 8.** Latitude–amplitude profiles of PW1 (left panel) and PW2 (right panel) in the Northern Hemisphere extratropics for each SPV cluster (C1–C10), shown in different colors. Cluster C1 (unstable vortex) is shown with a dotted line, clusters C2–C6 (single vortex) with solid lines, and clusters C7–C10 (split vortex) with dashed lines. Wave amplitudes are computed from daily 10 hPa GPH fields and averaged over all days classified into each respective cluster during the extended winter season (NDJFM), across all experiments. The black lines in both panels represent the climatological PW1 and PW2 amplitudes calculated from the control experiment. The legend for both panels is shown in the right panel. Note the different scaling in both panels.

configurations. In particular, although C3 has low PW2 amplitude, its vortex remains strongly deformed due to dominant PW1 activity, giving it an oval shape.

Figure 9 presents the extended winter occurrence frequencies of SPV clusters C1–C10 (as shown in Figure 7) in the control run on top of each panel, along with the occurrence differences in the three sensitivity experiments (EA–C, NA–C, HI–C) as color bars. The occurrence frequencies are computed using all ensemble members, with standard deviations calculated across the respective six ensembles. For the control run, these standard deviations are reported as $\pm$ values, while in the sensitivity experiments, they are shown as error bars on the bars representing mean differences. Hatching indicates that the sign of the anomaly is consistent across at least five out of six ensemble members.

Among the clusters, C2 has the highest mean occurrence in the control run ($35.12\% \pm 0.91\%$), followed by C6 ($30.21\% \pm 2.10\%$), C3 ($22.04\% \pm 1.80\%$), and C5 ($9.39\% \pm 1.41\%$). While the absolute frequencies vary under different experiments, these four clusters remain the most frequent across all experiments. Among the less frequent clusters, C4 represents displaced SSW states, whereas C7-C10 correspond to split SSWs. C1, in contrast, does not exhibit a well-defined vortex structure and represents unstable vortex configurations.

The HI experiment shows a consistent increase in the frequency of C1, C2, and C5, and a consistent decrease in C6 and C9. NA shows consistent increases in C1, C4, C5, and C8. The EA experiment shows consistent increases in C2, C4, C5, and C8, and consistent decreases in C6 and C9. Cluster C3 exhibits a mean decrease in all experiments but does not show ensemble consistency in any case.





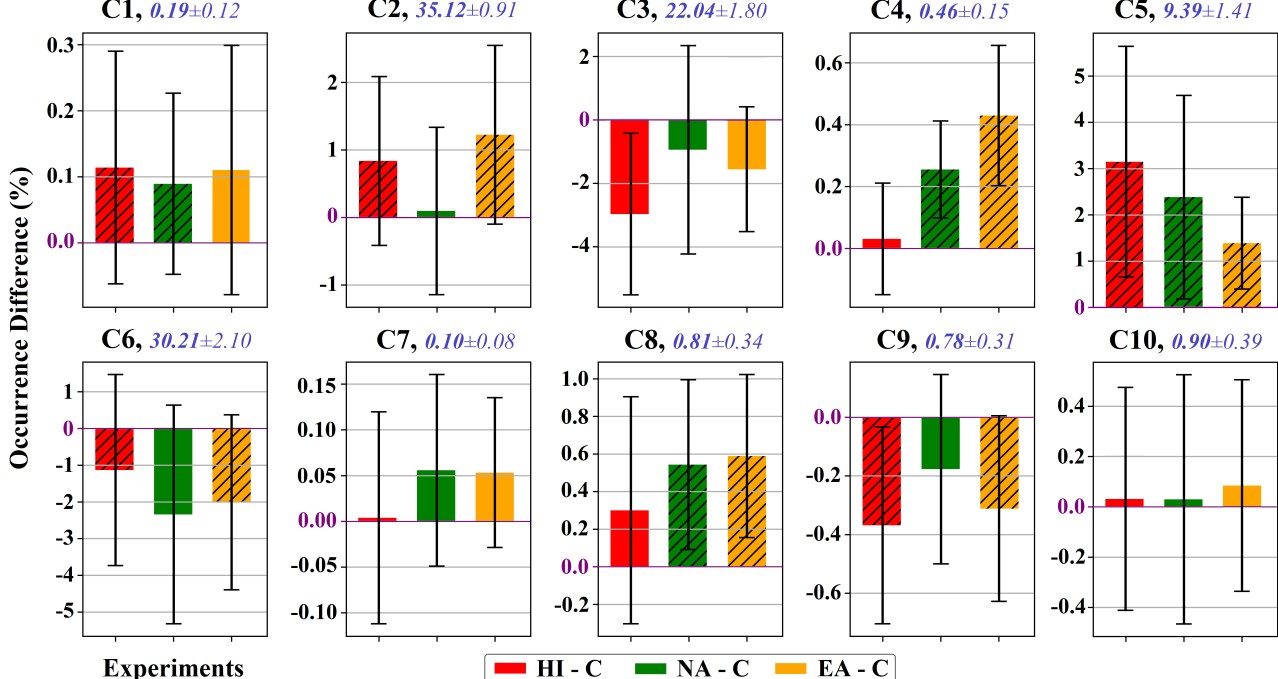

**Figure 9.** Occurrence frequency differences (%) between each experiment and the control run for SPV clusters C1–C10 during the extended winter season (NDJFM). Differences are calculated as the mean difference across ensemble members for each experiment (HI–C: red, NA–C: green, EA–C: orange). Error bars represent the standard deviation of the differences across the six ensemble members. The horizontal purple line at 0% marks the control run baseline. Bars are hatched when the sign of the difference is consistent with the ensemble-mean value in at least five out of six ensemble members. The extended winter occurrence frequency (%) of each cluster in the control run is shown above each panel in blue, with ± values indicating the standard deviation across control ensemble members. Note the different y-axis scaling for each cluster.

## 3.3 HI

Figure 10 presents the contributions of individual SPV clusters (C1–C10) to the 10 hPa GPH anomaly in the HI experiment, as shown in Figure 3b. The sum of these class contributions reconstructs the full climatological GPH anomaly in HI relative to the control run (Mehrdad et al., 2024). As described in Section 2.4, each cluster contributes through two components: WCVC, which reflects changes in the cluster's mean structure between the control and the sensitivity experiment; and FSDC, which captures the impact of changes in cluster occurrence frequency.

The clusters do not contribute equally; their impacts vary in spatial pattern, magnitude, and consistency. For instance, C6 contributes to the anomaly through its WCVC component by consistently deepening the vortex over the central Arctic (Figure 10 panel C6a). This indicates that when C6 occurs in the HI experiment, it consistently exhibits a deeper vortex than in the control run. C5, on the other hand, contributes via its FSDC component (Figure 10 panel C5b), displaying a pattern resembling





a weakened PW1 structure. This reflects the impact of a consistent increase in C5's occurrence frequency in HI (see Figure 9) on the GPH field.

Other notable contributors include C2 and C3, which contribute consistently through their WCVC components (Figure 10 panels C2a and C3a). Both show patterns opposite to the PW1 climatology, implying that their internal changes weaken the PW1-like structure. In C2, the WCVC component is the main contributor to the lower GPH observed over the Pacific sector. Other clusters contribute more modestly; for instance, the consistent reduction in C9's occurrence frequency in HI leads to a modest but consistent negative GPH anomaly at high latitudes via the FSDC component (Figure 10 panel C9b).

Figure 11 shows the class contributions to the zonal-mean zonal wind anomaly in the HI experiment, as shown in Figure 6a. In accordance with the analysis of the GPH10 class contributions, the direct effects of the imposed forcing are most apparent in the WCVC components of C2, C3, C5, and C6 (the most frequent classes), which exhibit weakening of the westerlies in the midlatitudes extending into the middle stratosphere. C6 is the primary contributor to the high-latitude westerly strengthening through its WCVC component (Figure 11 panel C6a). The consistent increases in the occurrence frequencies of C5 and C9 (see Figure 9) also result in modest but consistent high-latitude westerly wind enhancements via their FSDC components (Figure 11 panels C5b and C9b). C3's FSDC component also contributes to enhanced westerlies in the upper stratosphere, although this contribution is not consistent across ensembles—reflecting the similarly inconsistent reduction in C3's occurrence in the HI experiment. Furthermore, the WCVC components of C2 and C3 (Figure 11 panels C2a and C3a) exhibit anomaly patterns resembling the overall HI zonal wind anomaly, reinforcing their role in shaping the response.

Figure 12 shows the class contributions to the EP flux divergence anomaly in the HI experiment, as presented in Figure 6d. The WCVC component of C2 is the primary contributor to the positive EP flux divergence anomalies in the midlatitude middle and upper stratosphere (Figure 12 panel C2a), indicating reduced wave drag in these regions. Clusters C3 and C6 also contribute consistently to these positive anomalies in the midlatitude middle stratosphere through their WCVC components, although more modestly (Figures 12 panels C3a and C6a). These two clusters are also the main contributors to the negative EP flux divergence anomalies in the high-latitude lower and middle stratosphere, implying locally enhanced wave drag. Additionally, the consistent increase in the frequency of C5 (see Figure 9) resulted in a modest but consistent positive contribution to EP flux divergence anomalies in the high-latitude upper stratosphere via its FSDC component (Figure 12C5b). C3's FSDC component (Figure 12 panel C3b) also contributes relatively strongly to positive anomalies in the high-latitude upper stratosphere, although this contribution is not consistent across ensembles—reflecting the inconsistent reduction in C3 occurrence in the HI experiment (Figure 9.

## 3.4 NA

Figure 13 shows the class contributions to the 10 hPa GPH anomaly in the NA experiment, as shown in Figure 3c. In general, both the class contributions and the resulting anomaly patterns are less consistent compared to those observed in the HI and EA (see below) experiments. However, several clusters do exhibit notable and consistent responses. The consistent increase in the frequency of C5 (see Figure 9) results in a significant and consistent contribution through its FSDC component, which resembles a weakening of the climatological PW1 structure (Figure 13 panel C5b). C2 also contributes consistently to the negative







**Figure 10.** Class contributions to the 10 hPa GPH anomaly for the HI experiment. Each panel (e.g., C1a/C1b to C10a/C10b) corresponds to one SPV class. For each class, the top sub-panel (a) shows the Within-Cluster Variability Contribution (WCVC), and the bottom sub-panel (b) shows the Frequency-weighted Seasonal Deviation Contribution (FSDC). For each SPV cluster, the mean boundaries shown in Figure 7 are also overlaid with cyan contours in the WCVC sub-panels. Dotted areas indicate regions where the class contribution is consistent across at least five out of six ensemble members. The green outline marks the forced region in the HI experiment.

GPH anomaly over the Pacific sector via its WCVC component (Figure 13 panel C2a). While the same WCVC component also contributes to a positive GPH anomaly over the Arctic, that signal is largely inconsistent across ensembles. Additionally, C6



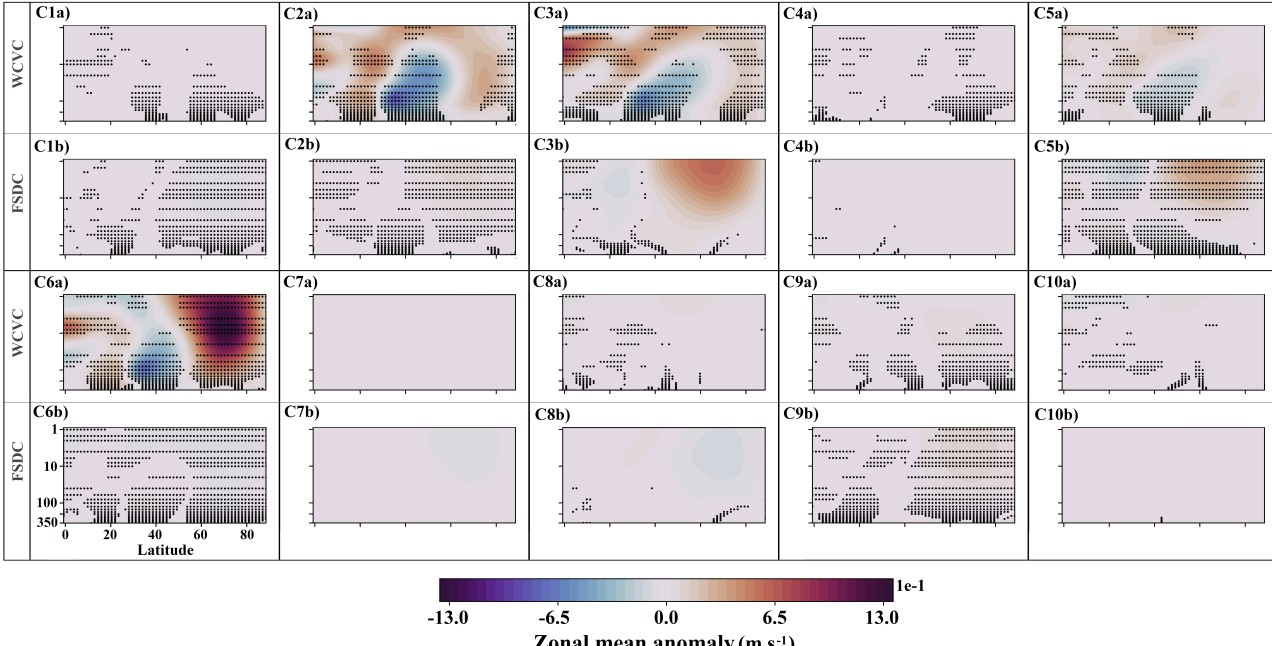

**Figure 11.** Similar to Figure 10 but computed for zonal-mean zonal wind. The vertical levels in latitude-height plots are represented in hPa.

contributes consistently to the positive GPH anomaly over northeastern Asia through its WCVC component (Figure 13 panel C6a). Modest but consistent contributions are also seen from C4 and C8, whose consistently increased frequencies in the NA experiment resulted in weak positive GPH anomalies over the Arctic via their FSDC components (Figure 13 panels C4b and C8b).

Figure 14 shows the class contributions to the zonal-mean zonal wind anomaly in the NA experiment, as presented in Figure 6b. The direct weakening effects of zonal-mean zonal wind westerlies associated with the imposed forcing are evident in the WCVC components of C2, C3, C5, and C6. The consistent increase in the occurrence frequency of C5 (see Figure 9) led to a contribution through its FSDC component with consistent strengthening of the zonal-mean zonal wind in the high-latitude upper stratosphere (Figure 14 panel C5b). Similarly, the consistent increased occurrence of C4 and C8 in NA resulted in a modest but consistent weakening of the westerlies in the high-latitude stratosphere through their FSDC components (Figures 14 panels C4b and C8b). Other notable contributions include the WCVC component of C2, which shows a strong but mostly inconsistent weakening of the westerlies across the mid- and high-latitude stratosphere (Figure 14 panel C2a). C6 also exhibits a strengthening effect at high latitudes upper stratosphere through its WCVC component, though this response is likewise inconsistent across ensemble members (Figure 14 panel C6a).

Figure 15 shows the class contributions to the EP flux divergence anomaly in the NA experiment, as presented in Figure 6e. The consistent increases in the occurrence frequencies of C4, C5, and C8 (see Figure 9) resulted in modest but consistent positive contributions to EP flux divergence in the high-latitude upper stratosphere via their FSDC components (Figure 15 panels



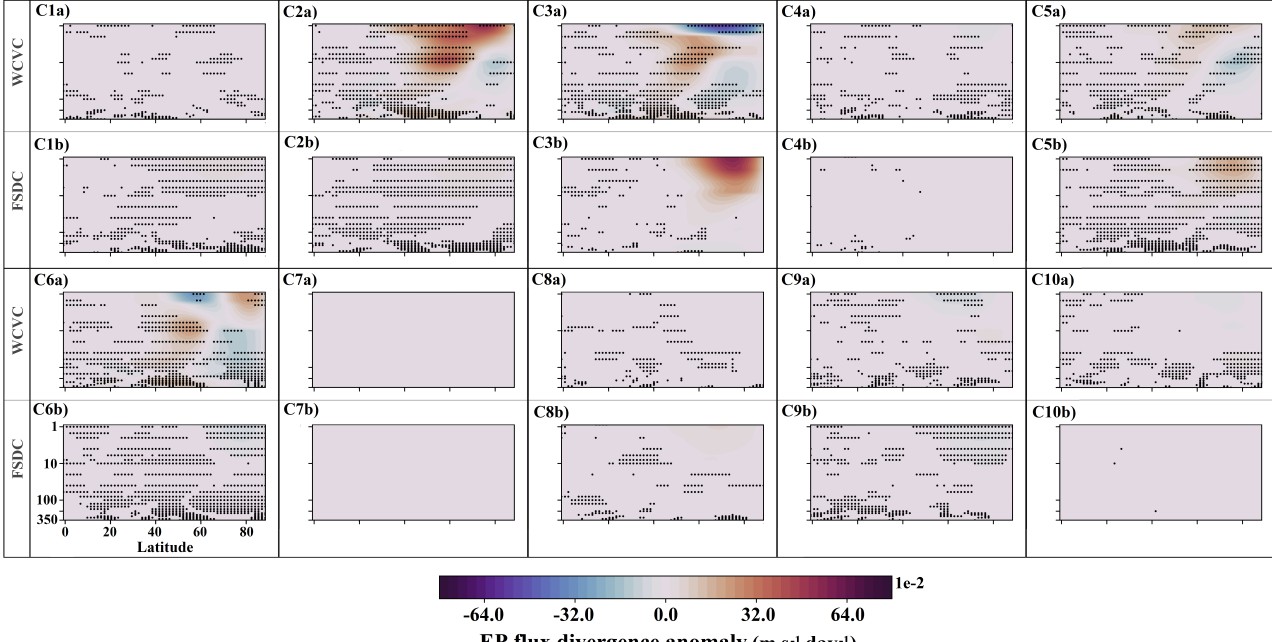

**Figure 12.** Similar to Figure 11 but computed for EP flux divergence.

C4b, C5b, and C8b). In addition, the WCVC component of C2 contributes consistently to more positive EP flux divergence anomalies in the high-latitude lower stratosphere, extending into the midlatitude upper stratosphere (Figure 15 panel C2a). Contributions from other clusters, while sometimes notable, are generally not consistent across ensemble members.

## 3.5 EA

Figure 16 shows the contributions of individual SPV clusters to the 10 hPa GPH anomaly in the EA experiment, as shown in Figure 3d. Overall, the cluster contributions in EA are similar to those in NA, but they are more consistent across ensemble members. The consistent increase in the occurrence frequency of C5 in EA (see Figure 9) resulted in an FSDC contribution that resembles a weakening of the climatological PW1 pattern (Figure 16 panel C5b). The consistent reduction in the occurrence frequency of C6 also contributes a wave-1-like pattern, though slightly shifted relative to the climatology (Figure 16 panel C6b). Additionally, the increased occurrence of C4 and C8 (clusters representing SSWs) led to consistent positive GPH anomalies at high latitudes through their FSDC components (Figure 16 panels C4b and C8b). The consistent decrease in C9 and increase in C2 occurrence frequencies both contribute to a moderate but consistent negative GPH anomaly over high latitudes (Figure 16 panels C9b and C2b).

The WCVC components of C2, C3, C5, and C6 also contribute notably to the total anomaly. C2 shows a dipole pattern with a negative anomaly over the Pacific sector and a positive anomaly over high latitudes (Figure 16 panel C2a). C3 exhibits a




**Figure 13.** Similar to Figure 10 but for the NA experiment.

moderate weakening of the PW1 structure (Figure 16 panel C3a), while C6 displays a mostly negative anomaly with a localized positive signal over northeastern Asia (Figure 16 panel C6a).

Figure 17 shows the class contributions to the zonal-mean zonal wind anomaly in the EA experiment, as presented in Figure 6c. The WCVC component of C2 is the primary and most consistent contributor to the weakening of high-latitude westerlies and the strengthening of midlatitude westerlies (Figure 17 panels C2a). C5 and C6 exhibit similar patterns through





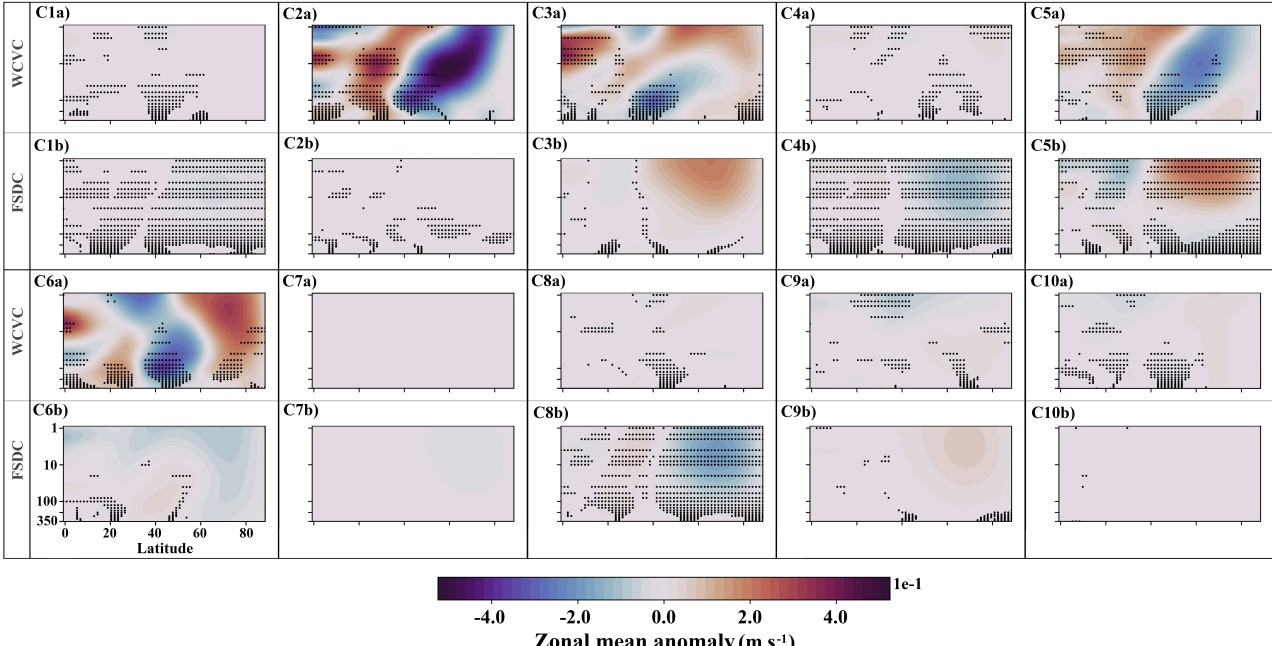

**Figure 14.** Similar to Figure 11 but for the NA experiment.

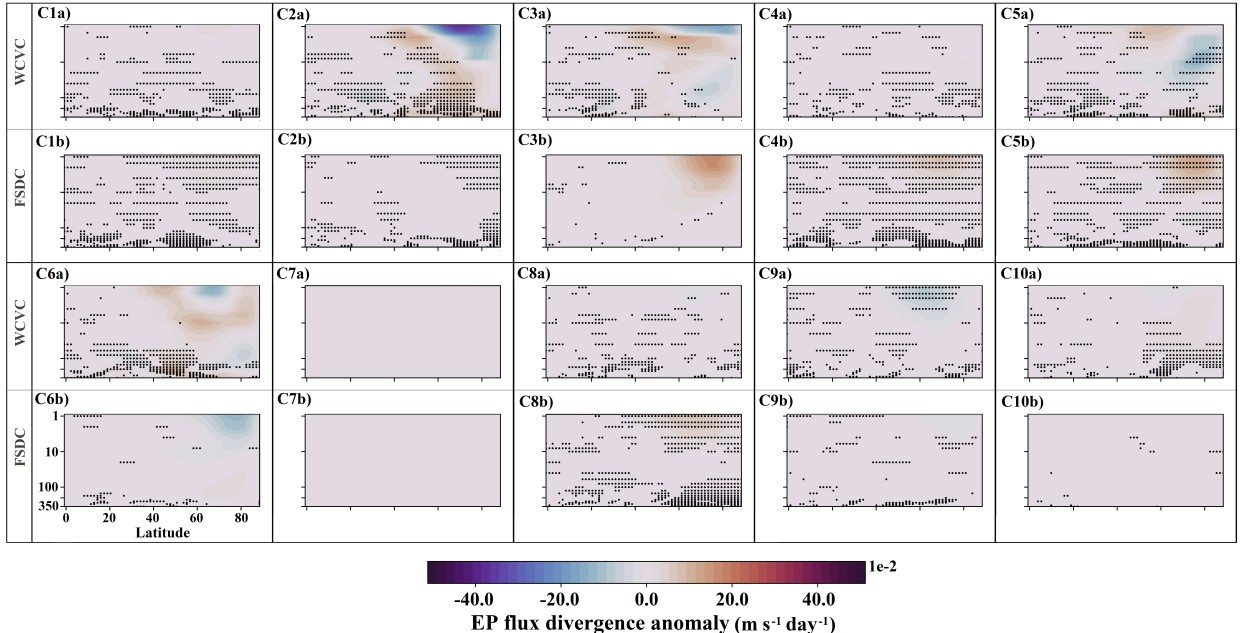

**Figure 15.** Similar to Figure 12 but for the NA experiment.





**Figure 16.** Similar to Figure 10 but for the EA experiment.

their WCVC components, but their contributions are more moderate and less consistent (Figure 17 panels C5a and C6a). The increased occurrence frequencies of C4 and C8 (see Figure 9) resulted in modest but consistent weakening of high-latitude westerlies via their FSDC components (Figure 17 panels C4b and C8b). Meanwhile, the increased frequencies of C2 and C5, along with the decreased frequency of C9, also led to consistent enhancements of high-latitude westerlies through





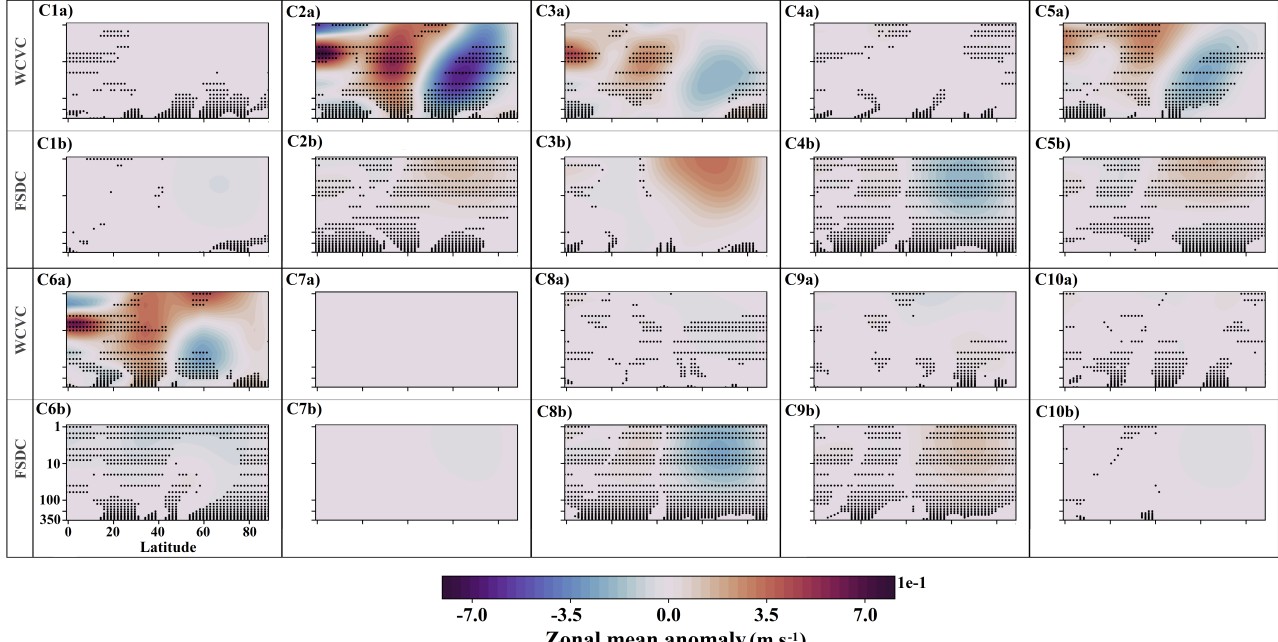

**Figure 17.** Similar to Figure 11 but for the EA experiment.

their respective FSDC contributions (Figure 17 panels C2b, C5b, and C9b). However, these contributions appear to act in

compensation to the net weakening of high-latitude westerlies observed in the overall anomaly (Figure 6c).

Figure 18 shows the class contributions to the EP flux divergence anomaly in the EA experiment, as presented in Figure 6f. The consistent increases in the occurrence frequencies of C2, C4, C5, and C8 (see Figure 9) resulted in modest but consistent positive contributions to EP flux divergence anomalies in the high-latitude upper stratosphere via their FSDC components (Figure 18 panels C2b, C4b, C5b, and C8b). Conversely, the consistent decrease in the occurrence frequency of C9 led to a

370 modest but consistent negative FSDC contribution to the anomaly (Figure 18 panel C9b). C5 through WCVC component is the primary contributor to the positive EP flux divergence anomalies in the high-latitude upper stratosphere (Figure 18 panel C5a). In addition, the WCVC components of C6, C3, and C2, listed in order of decreasing strength, also contribute consistently to positive EP flux divergence anomalies in the middle and upper stratosphere (Figure 18 panels C6a, C3a, and C2a).

## 4 Discussion and conclusions

This study provides new insights into the response of the SPV to gravity wave forcing in Northern Hemisphere stratospheric hotspots using a novel shape-based clustering approach. The SPV does not respond in a zonally symmetric manner to localized gravity wave forcing in the defined hotspot regions. Across all experiments, regardless of the specific hotspot location, the response consistently featured a PW1-like displacement of the vortex core (Figure 3). This involved a more irregular, ragged



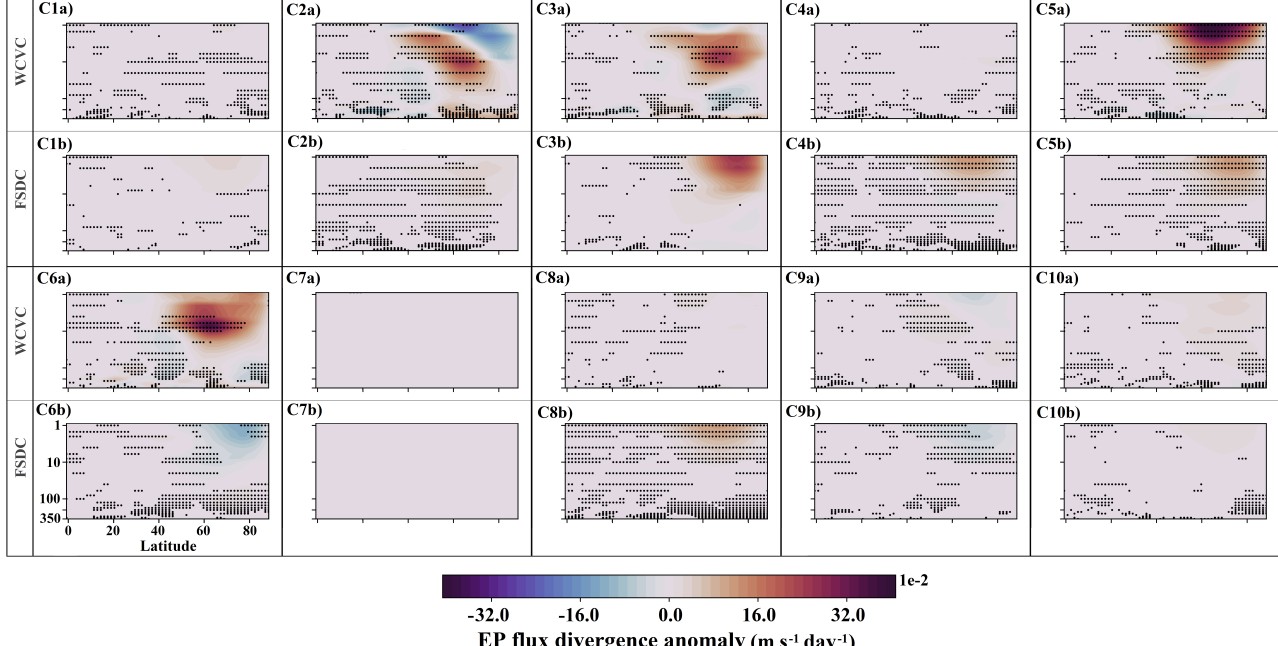

**Figure 18.** Similar to Figure 12 but for the EA experiment.

edge over northern Eurasia, although the detailed anomaly pattern remains distinct for each case. The dominant dynamical
feature underlying these anomalies was the consistent reduction in the amplitude of PW1 across all experiments. This is also
evident in the EP flux divergence anomalies (Figure 6), which showed positive EP flux divergence anomalies associated with
the PW1 structure in the midlatitude stratosphere Mehrdad et al. (2025a).

These anomalies manifested as negative GPH anomalies within the forced region and positive anomalies northward of it
(Figure 3). A plausible explanation is that enhanced GW drag near the forced region disrupts the SPV edge through increased
small-scale wave breaking and secondary instabilities (Coy et al., 2024). This mixing across the vortex boundary locally blurs
the typically sharp SPV edge, resulting in the observed pattern of anomalies. Specifically, positive GPH anomalies at the SPV
periphery often indicate incursions of low-PV, warm high-pressure ridges, while negative anomalies reflect high-PV polar air
being displaced equatorward, forming cold low-pressure troughs (Figure A1). Such patterns are characteristic of surf zone
wave activity impinging on the SPV (Waugh, 1997). The HI experiment, which represents the southernmost hotspot region,
showed the strongest response, with the positive anomaly band north of the forcing extending only to the northern Eurasian
coastline. This configuration effectively sharpens the vortex edge in that sector and further deepens the polar vortex core toward
higher latitudes.

Examining the forcing effect on the occurrence frequency of the preferred SPV geometry clusters (Figure 7) revealed a
consistent increase in the occurrence of C5 across all experiments (Figure 9). Among the single vortex clusters (C2–C6), C5 had
the weakest PW1 amplitude except for C4, which represents displaced SSW events (Figure 8, left panel). The PW1 amplitude



associated with C5 remained well below the climatological mean, so its increased occurrence frequency contributed directly to the reduction in PW1 amplitude observed in all three experiments. This effect is clearly evident in the FSDC component of C5's contribution to the 10 hPa GPH anomaly, which exhibited a pattern opposite to the PW1 climatology (panel C5b in Figures 10, 13, and 16). A similar signal is apparent in the EP flux divergence anomalies, with positive anomalies consistent

with a weakened PW1 structure (panel C5b in Figures 12, 15, and 18) Mehrdad et al. (2025a). Although the amplitude of this contribution varied among experiments, it remained a consistent feature. However, it did not represent the dominant driver of the anomalies in any individual case.

C5 is the only cluster that showed a consistent and similar sign change in occurrence frequency across all experiments. There were other clusters where the occurrence frequency change was only consistent in one or two experiments, such as C4 and

405 C8, displaying consistent increases in occurrence frequency only in the EA and NA experiments. These clusters are associated with SSW events, and their increased occurrence contributed to the anomalies by weakening the SPV (see panels C4b and C8b in Figures 16 and 17 for EA, as well as Figures 13 and 14 for NA). However, because these clusters have relatively low occurrence frequencies in the climatology, their contributions, though consistent, remained comparatively weak. The robust increased occurrence of these SSW clusters in the EA experiment is consistent with the higher number of SSW events per

410 decade diagnosed for this case (Mehrdad et al., 2025a).

The WCVC component of the class contribution quantifies how the forcing influences the general structure and geometry of a given cluster, but these structural changes are typically small enough that they do not lead to reassignment to a different cluster (Mehrdad et al., 2024). The WCVC contribution was generally stronger in the most frequent clusters (C2, C3, C5, and C6) and often represented the dominant pathway shaping the observed anomalies. However, these structural responses did not

always occur consistently across ensemble members, underscoring the role of internal variability, a feature particularly evident in the NA experiment. When consistent, however, the WCVC patterns could be useful for revealing the underlying mechanisms and physically meaningful signals by which localized forcing modifies the SPV.

In the HI experiment, the WCVC components of C2 and C6 were the main contributors modulating the 10 hPa GPH anomalies. C2 contributed to a PW1-like pattern, especially in the midlatitudes, while C6 primarily contributed to the lower GPH

values observed in the polar region (see panels C2a and C6a of Figure 10). The WCVC component of C3 also contributed with a PW1-like pattern; together with C2, these two clusters, which have strong PW1 amplitudes (left panel of Figure 8), contributed to the anomaly with a pattern opposite to the PW1 climatology. This suggests that the forcing tends to weaken the PW1 amplitude in these clusters that normally exhibit a strong PW1 signal.

The WCVC components of C2, C3, and C6 consistently generated positive GPH anomalies north of the forced region

and negative anomalies within it. This pattern aligned zonally along the SPV boundaries, which correspond to the GPH isobars representing each cluster's mean circulation, highlighting how the forcing interacts with the geometry of the SPV. The resulting positive anomaly patches reflected localized SPV edge mixing driven by the forcing and tended to follow the mean circulation associated with each cluster. Notably, for C6, this interaction strengthened the SPV edge by producing a narrow band of higher GPH north of the forced region. This demonstrates how the relative position and geometry of the SPV relative to the hotspot can

determine whether the hotspot forcing leads to mixing and weakening or, conversely, to intensification of the vortex structure.





The deceleration of the zonal mean zonal wind near the forcing region in HI was evident in the WCVC components of the most frequent clusters (Figure 11), indicating the direct and consistent role of the imposed forcing in shaping these contributions. The WCVC component of C6 emerged as the main contributor to the zonal wind acceleration over the polar regions, consistent with its role in deepening the SPV core. Comparing the class contribution to the EP flux divergence (Figure 12) and zonal-mean zonal wind revealed that they did not always align straightforwardly across clusters. For example, in the midlatitude lower and mid-stratosphere, the deceleration of the mean flow was not accompanied by corresponding negative anomalies in EP flux divergence as would typically be expected. This discrepancy suggested that other momentum sources, such as GW drag imposed in the experiment Mehrdad et al. (2025a), play a dominant role in this region. Similar inconsistencies were apparent in the C6 WCVC component, where positive anomalies in EP flux divergence did not coincide with stronger westerlies (see panel C6a in Figures 11 and 12). Overall, such differences arose because the class contribution framework captures the simultaneous response but does not account for the delayed adjustments that are typical of wave–mean flow interactions.

For the NA experiment, the most consistent contributions arose from the FSDC components of clusters C4, C5, and C9, which reflect the consistent increase in their occurrence frequencies. In contrast, other class contributions, while locally strong, tended to be less consistent. The WCVC components of C2, C3, and C6, for example, displayed localized patches of positive GPH anomalies in the northern part of the forced region, extending along the SPV boundary in a manner similar to the HI experiment (Figure 13), but with only partial consistency. Notably, the WCVC contribution of C2 was particularly strong and extends into the polar region, but it was consistent only in a relatively small area northeast of North America. This pattern was linked to the C2 WCVC's associated deceleration of the zonal mean zonal wind in the mid- and high-latitude stratosphere (panel C2a in Figure 14), which also showed patchy consistency, especially at higher latitudes. Overall, this highlights how internal variability in the NA experiment tended to obscure clear, spatially coherent responses to the imposed forcing, in contrast to the more robust signals evident in the HI experiment.

In the EA experiment, the class contributions were generally similar to those in the NA experiment but exhibited more spatially consistent signals. This similarity likely stemmed from the more northward placement of the NA and EA hotspots compared to the HI hotspot, which might influence the location and magnitude of the SPV response. The greater consistency observed in the EA experiment compared to NA could also be related to the relative phase alignment of the hotspot region with the stationary PW1 pattern (Mehrdad et al., 2025a). Across all experiments, a common feature was the presence of a positive GPH anomaly north of the forced region, indicative of PV mixing along the SPV edge. Notably, the responses in NA and EA, the two more poleward hotspots, showed deeper intrusions of this anomaly into the polar vortex core compared to the HI experiment.

Overall, localized gravity wave drag exerts a decisive yet geometry-sensitive influence on SPV variability. The shape-based clustering and class contribution framework developed in this study help reveal robust and spatially coherent responses to localized forcing that might otherwise be obscured by internal variability within the full ensemble. While the current analysis is limited to present-day climatological conditions, extending this morphology-based approach to transient climate scenarios and observational reanalyses would further test its utility. Such applications could improve our understanding of the mechanisms governing SPV variability and offer new opportunities for enhancing subseasonal prediction.



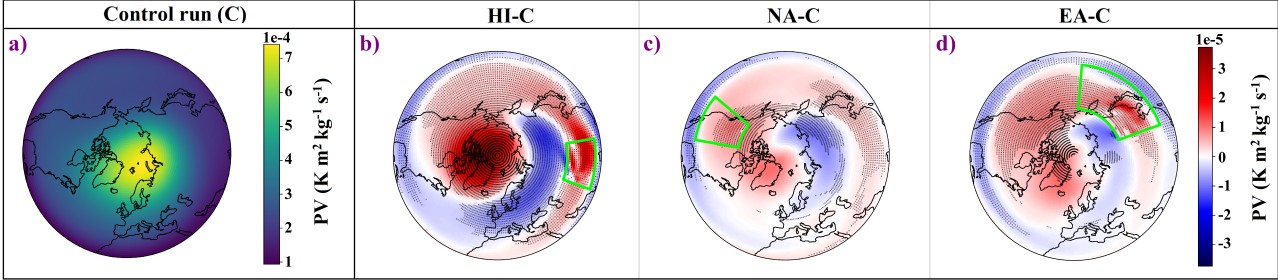

**Figure A1.** Same as Figure 3, but for PV on the 850 K isentropic surface. Panel (a) shows the PV climatology from the control run. Panels (b–d) display PV anomalies for the HI, NA, and EA simulations, respectively, relative to the control run. Dotted areas indicate regions where anomalies are consistent across at least five of six ensemble members. The forced regions are outlined in green.

*Code and data availability.* The daily output datasets for the Control run and the HI, NA, and EA sensitivity experiments used in this study are publicly available through the World Data Center for Climate (WDCC) at DKRZ under the project CC-LGWF (https://www.wdc-climate. de/ui/project?acronym=CC-LGWF). The corresponding DOIs are: https://doi.org/10.26050/WDCC/UAICON_GW_C for all six ensemble members of the Control run (Mehrdad et al., 2025b), https://doi.org/10.26050/WDCC/UAICON_GW_HI for HI (Mehrdad et al., 2025d),
https://doi.org/10.26050/WDCC/UAICON_GW_NA for NA (Mehrdad et al., 2025e), and https://doi.org/10.26050/WDCC/UAICON_GW_ EA for EA (Mehrdad et al., 2025c) experiments. The same dataset has also been used in Mehrdad et al. (2025a).

## Appendix A: Potential vorticity (PV)

Figure A1a shows the climatology of PV on the 850 K isentropic surface, which approximately corresponds to the 10 hPa pressure level, for the control simulation. Panels A1b–d display the corresponding PV anomalies for the HI, NA, and EA
experiments. Both the climatological structure and anomaly patterns closely resemble those observed in the 10 hPa GPH fields presented in Figure 3. As expected, the PV fields exhibit an inverse relationship with the GPH fields; that is, regions of high PV generally correspond to low GPH.

*Author contributions.* The study was originally conceptualized by Sina Mehrdad (S.M.) and Christoph Jacobi (C.J.), with substantial input from all co-authors. Model simulations were designed and performed by S.M. and Sajedeh Marjani (S.Ma.), under the supervision of C.J.
The manuscript was written and prepared by S.M., with additional development and editorial input from C.J. All authors contributed to discussions throughout the project, provided critical feedback, and reviewed the final manuscript.

*Competing interests.* The authors declare that no competing interests are present.



*Acknowledgements.* We gratefully acknowledge the funding by the Deutsche Forschungsgemeinschaft (DFG, German Research Foundation) – Projektnummer 268020496 – TRR 172, within the Transregional Collaborative Research Center "ArctiC Amplification: Climate Relevant Atmospheric and SurfaCe Processes, and Feedback Mechanisms (AC)[3]". This work used resources of the Deutsches Klimarechenzentrum (DKRZ) granted by its Scientific Steering Committee (WLA) under project IDs bb1238 and bb1438.



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
