# Peer review of "Northern Hemisphere Stratospheric Polar Vortex Morphology under Localized Gravity Wave Forcing: A Shape-Based Classification"

_EGUsphere, 2025_

## Author Comment (AC1)

We thank the referee for the insightful comments. Below, we repeat the reviewer's remarks in red italics, and add our respective responses in normal text.

**Reviewer 1**

**General comments**

This study assesses the Northern Hemisphere stratospheric polar vortex response to localized gravity wave forcing above three hotspot regions, the Himalayas, Northwest America, and East Asia, using UA-ICON GCM. The results highlight that all hotspot forcings consistently reduce planetary wave 1 amplitude, which is discussed in detail. I find the study highly relevant, especially due to the classification framework developed and its application to transient climate simulations and reanalyses. I recommend publication once the minor comments below are addressed.

We sincerely thank the reviewer for the thoughtful evaluation and positive feedback on our study. We appreciate the reviewer's valuable comments, which have helped improve the scientific clarity and overall quality of the manuscript.

**Specific comments**

I noticed that in Fig. 6 in Mehrdad (2025a), the zonal-mean climatology of the tendencies induced by the OGW parameterization scheme shows a secondary maximum in the lower stratosphere over midlatitudes, within the so-called valve layer (Kruse et al., 2016). However, this maximum is located around and below 100 hPa. This contrasts with the breaking of freely propagating OGWs above the center of the UTLS jet starting rather above in CMIP6 AMIP simulations (Hajkova and Sacha, 2024). Can you comment on this deficiency or model tuning with respect to vertical profiles of OGWD in the sensitivity simulations above NA and HI, where the breaking is maximized below 100 hPa (see Fig. 3 in Mehrdad (2025a))? This can consequently affect the polar vortex response simulated by UA-ICON.

We thank the reviewer for the insightful comment emphasizing the importance of the model's climatological vertical distribution of subgrid-scale orographic (SSO) gravity-wave (GW) drag in the context of our sensitivity experiments.

UA-ICON employs the SSO GW drag scheme of Lott and Miller (1997) using its default parameter settings. Figure 1 (adapted from Figure 6 in Mehrdad et al., 2025a, also added as Figure A2a in the revised manuscript) shows the extended-winter (NDJFM) zonal-mean climatology of the SSO-induced zonal-wind tendency together with the zonal-mean wind. Two maxima appear: (i) a primary one in the upper stratosphere and (ii) a secondary maximum in the mid-latitude lower stratosphere, centred slightly below 100 hPa, corresponding to the so-called valve layer (Kruse et al., 2016). This maximum is located on the upper flank of the zonal-mean UTLS jet. A comparable structure appears in other UA-ICON climatologies (e.g., Kunze et al., 2025) using the same scheme and parameters. The precise altitude of this secondary maximum is sensitive to both the local jet structure and to scheme parameters such as the critical Froude and Richardson numbers.

In our simulations, the secondary maximum lies near the lower boundary of the valve layer, somewhat lower than the multi-model mean of CMIP6 AMIP OGWD climatologies reported by Hájková & Šácha (2024), where freely propagating OGWs typically deposit momentum above  $\approx 100$  hPa. However, their analysis also shows a large inter-model spread in the vertical position of this peak, linked to scheme design and tuning. Thus, the placement in UA-ICON remains physically reasonable and consistent with the broader range found in CMIP6.

Figure 2 (adapted from Figure 3 in Mehrdad et al., 2025a) further illustrates that the vertical structure of SSO drag differs across the EA, NA, and HI hotspot regions. In the control climatology (blue curves), NA and HI exhibit enhanced drag centered below ≈ 100 hPa, whereas EA shows a maximum extending higher into the lower stratosphere. These regional variations arise from differences in GW generated in the SSO scheme and in local jet altitude, both of which control where critical-level saturation occurs. When the forcing is intensified (orange/green/red curves), the vertical distribution responds accordingly—strengthening the existing lower-stratospheric peak over NA and HI but the upper peak over EA, demonstrating the role of the background climatology in constraining the response.

Regarding the possible implications for the Northern Hemisphere stratospheric polar vortex (SPV), Hájková & Šácha (2024) found that stronger SSO GW drag in the valve layer tends to enhance resolved-wave drag (EPFD) in the polar lower stratosphere through refractive-index changes ("amplifying interaction"). However, the inter-model mean zonal-wind differences do not show a robust relationship with either the valve-layer SSO GW drag or EPFD. Consequently, while a slightly lower valve-layer peak in UA-ICON could influence the details of upward wave propagation near the jet, its effect on the climatological SPV strength is expected to be minor within the range of uncertainty documented by CMIP6. A dedicated sensitivity test varying the SSO-scheme parameters would be required to quantify this influence, which lies beyond the present study.

We have added Appendix A (lines 486–506), which provides a concise summary of the spatial and vertical distribution of the imposed SSO-induced zonal wind tendencies by hotspot. We also include the zonal-mean zonal wind tendency climatology of the model control run (Figure A2a of the revised manuscript) and briefly note the lower-stratospheric valve-layer maximum in our UA-ICON simulation. This appendix supplies the essential vertical context without duplicating Mehrdad et al. (2025a).

In addition, we added a paragraph in Section 4.4 (Model limitations and implications) of the revised manuscript (lines 465-472) to explicitly discuss the model's SSO GW drag climatology, its relationship to the valve layer, and how the vertical placement compares with the CMIP6 AMIP multi-model mean reported by Hájková & Šácha (2024).

Figure 1: Zonal-mean climatology of the SSO-induced zonal wind tendency (color shading), with red contours indicating the climatology of the zonal-mean zonal wind (m s-1) in the control simulation. The climatologies are computed for the extended winter period (NDJFM). Adapted from Mehrdad et al. (2025a).

Figure 2: (a–c) Maps showing the latitudinal bands of the forcing regions for the EA (a), NA (b), and HI (c) hotspot. The "in" regions correspond to the areas where the forcing is applied, while the "out" regions represent the same latitude bands outside the hotspot regions. (d–f) Vertical profiles of the SSO-induced zonal wind tendency (m s-2) during the extended winter period (NDJFM), calculated from ensemble means. Each column corresponds to one sensitivity simulation: EA (d), NA (e), and HI (f). The solid blue lines show the control run climatology, computed as the difference between the "in" and "out" regions. Solid orange, green, and red lines represent the corresponding differences between "in" and "out" regions in the EA, NA, and HI simulations, respectively. Dashed lines indicate the difference between each sensitivity simulation and the control run for the forced ("in") regions. Thickened line segments denote regions where the differences are consistent.

I miss the motivation why such a methodology has been applied to classify the SPV geometry compared to either standard clustering techniques (e.g. k-means in Kretschmer et al (2018)) and/or standard techniques for split and displacement identification (e.g. Seviour et al, 2013)

Thank you for this helpful suggestion. We have revised Section 2.3 (Classification of SPV geometry) to make the motivation explicit. In brief, previous work has used geometric diagnostics (e.g., centroid latitude, aspect ratio) to identify displaced/split states (Seviour et al., 2013; Mitchell et al., 2013), which is transparent but coarse; or (ii) applied field-based

clustering on dynamical fields (e.g., k-means/hierarchical clustering as in Kretschmer et al., 2018), which offers a holistic view but is sensitive to small spatial shifts/rotations of the vortex. Our revised Section 2.3 explains that we combine the strengths of both by representing the vortex boundary geometry with Fourier descriptors and then clustering in that low-dimensional, physically interpretable feature space. Low-order harmonics quantify circularity/elongation/orientation, higher orders capture finer deformations, and truncation to the leading modes emphasizes large-scale morphology and reduces sensitivity to small-scale noise and minor spatial shifts. This provides a stable, morphology-based classification that is more informative than a handful of scalars yet less shift-sensitive than gridpoint-wise clustering. (See Section 2.3, lines 112-133)

**Can you include the value of the 18% threshold mentioned in Section 2.3.1 and Fig. 1?**

We thank the reviewer for this comment. The 18% threshold is defined as the 18th percentile of the daily 10 hPa geopotential height (GPH) distribution. We have revised the corresponding sentence in Section 2.3.1 (lines 137-138) to make it clearer.

Due to the extensiveness and unique methodology of the study, I think the whole community would appreciate an adoption of Open Science approaches to allow reproducing the extensive analysis in this study (e.g. Laken, 2016). In particular, I would recommend any kind of willingness of the authors to publish the code or a series of functions allowing to reproduce the figures in the paper. There are multiple ways to proceed, either to allow access upon request or via portals that allow assigning Digital Object Identifier (DOI) to the research outputs, e.g. ZENODO. I think it could enhance the quality and reliability of this publication.

We thank the reviewer for this valuable suggestion. The datasets used in this study are already publicly available via the WDCC. In addition, we now explicitly state in the "Data Availability" section that the analysis code used for the clustering, projection, and wave-diagnostic calculations is available upon request (lines 484-485 of the revised manuscript).

As shown in Mitchell et al (2011), splits are also accompanied by equatorward shift of the vortex (diagnosed by centroid latitude), i.e. a PW1-like pattern. In this view, I would suggest discussing results in Sections 3 and 4. Some studies find little (e.g. Maycock, A. C., and P. Hitchcock, 2015) or strong (e.g. Mitchell e al, 2013) differences between the surface impacts of split and displacement events. Have authors found any surface signatures in the sensitivity experiments?

We thank the reviewer for this helpful suggestion. To clarify the role of PW1 and PW2 in shaping split-vortex states, we have added a sentence on line 526-529 in the revised manuscript in (Appendix D of the revised manuscript).

Although we did not explicitly diagnose centroid latitude, the concurrent PW1 amplitude is qualitatively consistent with Mitchell et al. (2011).

Regarding the surface response, assessing the delayed tropospheric impacts of these stratospheric perturbations lies beyond the scope of the present cluster-based framework, which focuses on the simultaneous upper-stratospheric response. Nevertheless, as shown in Mehrdad et al. (2025a, their Appendix A/Figure A1), all sensitivity experiments display a tendency toward a negative Arctic Oscillation phase following the imposed gravity-wave hotspots, even though the SSW frequency is not significantly different among experiments (see Figure 4 in Mehrdad et al. 2025a). All experiment data are publicly available to facilitate further investigation of stratosphere—troposphere coupling and surface impacts. We consider this an interesting avenue for follow-up analysis and welcome such studies using our dataset.

I had trouble seeing dotted regions and contours in Figs. 3,4,5. I encourage authors to enhance their clarity. The choice of colours (cyan and green) in Fig. 10 could also be improved.

We thank the reviewer for this valuable feedback. In the revised manuscript, Figures 3 and 4 have been modified, and their subplots have been slightly enlarged to improve the visibility of the dotted regions and contours. The former Figure 5 (now Figure C1) is also scaled up. Regarding the cyan and lime contour overlays used, we carefully evaluated alternative colour options. However, these colours offer the best balance: they remain visible against both strong positive and negative anomalies, where darker contour colours tend to become indistinguishable. Although cyan and lime contours are less distinct over near-zero (white) regions, they provide better overall contrast across the full anomaly range. For this reason, we decided to retain these colours in the revised figures.

**I would replace abbreviations (EA, HI, NA) with their full length in subsection titles.**

We thank the reviewer for this helpful suggestion. The subsection titles have been revised to include the full names, now reading:

Himalayas (HI), Northwest America (NA), and East Asia (EA).

**I would move Fig. 5 to the appendix/supplement.**

We thank the reviewer for this helpful suggestion. In the revised manuscript, the former Figure 5 has been moved to Appendix C (lines 513–518) and is now labelled as Figure C1. The corresponding descriptive text has also been moved to this appendix. In addition, a reference to Appendix C has been added in Section 3.1 (line 241) to guide the reader to the section.

Have you considered decomposing EPFD into leading zonal planetary wave modes? As shown in Sacha et al (2021), diverse dynamical responses to OGWD hotspots, particularly given the different wavenumbers. This has been in details discussed in Kuchar et al (2022), highlighting that strong and intermittent OGW drag events above the Himalayas in the lower stratosphere are associated with anomalously increased upward RW propagation in the stratosphere. This is somewhat different to the

conclusion of this study. Overall, I suggest discusses differences in findings from previous studies in the manuscript.

Thank you. Yes, we have performed the wavenumber decomposition in our previous paper, Mehrdad et al. (2025a), where the UA-ICON ensemble experiments with regionally enhanced SSO GW drag showed that the resolved response is dominated by PW1, with suppressed upward and equatorward propagation and weakened wave drag overall (see Figures 10–12 in Mehrdad et al., 2025a).

In the present manuscript, we quantify the leading PW modes explicitly in terms of wave amplitude. Figure 4 shows that all three hotspot forcings reduce PW1 amplitudes at 10 hPa, together with a PW1-like displacement of the vortex (and smaller PW2 effects).

Regarding consistency with Šácha et al. (2021) and Kuchar et al. (2022): those studies diagnose short-time-scale, event-composite responses to naturally occurring strong orographic GW drag peaks near the valve layer in a specified-dynamics Canadian Middle Atmosphere Model (CMAM) setup. By contrast, our UA-ICON experiments apply regionally localized intensification of SSO drag and evaluate the NDJFM ensemble-mean response across 180 model years. In that climate-mean framework, the net signal is a hemispheric reduction of PW1 amplitude and weaker resolved wave drag in mid- and high latitudes, i.e., the time-integrated effect of many intermittent events together with background-state adjustments. This reconciles the different conclusions: short-lived event composites can show episodic responses that average out in the long-term response, leaving the dominant PW1 weakening documented here.

To address your suggestion directly, we have (i) clarified in the manuscript that PW1 dominates the response, with a pointer to our earlier study (lines 69-71 in the introduction, lines 246-247 in Section 3.1), and (ii) added a paragraph in the Discussion contrasting our climate-mean setup with the event-based analyses of Šácha (2021) and Kuchar (2022) (section 4.2 Common response to localized GW forcing, lines 376-383).

**Technical comments**

*L315 (Figure 9. -> (Figure 9).*

Thanks, The typo has been addressed in the revised manuscript.

**References**

Hajková, D., Sacha, P. Parameterized orographic gravity wave drag and dynamical effects in CMIP6 models. Clim Dyn 62, 2259–2284 (2024). https://doi.org/10.1007/s00382-023-07021-0

Kretschmer, M., D. Coumou, L. Agel, M. Barlow, E. Tziperman, and J. Cohen, 2018: More-Persistent Weak Stratospheric Polar Vortex States Linked to Cold Extremes. Bull. Amer. Meteor. Soc., 99, 49–60, https://doi.org/10.1175/BAMS-D-16-0259.1.

- Kruse, C. G., Smith, R. B., and Eckermann, S. D.: The midlatitude lower-stratospheric mountain wave "valve layer", Journal of the Atmospheric Sciences, 73, 5081–5100, https://doi.org/10.1175/JAS-D-16-0173.1, 2016.
- Kuchar, A., Sacha, P., Eichinger, R., Jacobi, C., Pisoft, P., and Rieder, H.: On the impact of Himalaya-induced gravity waves on the polar vortex, Rossby wave activity and ozone, EGUsphere [preprint], https://doi.org/10.5194/egusphere-2022-474, 2022.
- Laken, B. A. (2016). Can Open Science save us from a solar-driven monsoon? Journal of Space Weather and Space Climate, 6, A11. http://doi.org/10.1051/swsc/2016005020.
- Maycock, A. C., and P. Hitchcock (2015), Do split and displacement sudden stratospheric warmings have different annular mode signatures?, Geophys. Res. Lett., 42, 10,943–10,951, doi:10.1002/2015GL066754.
- Mehrdad, S., Marjani, S., Handorf, D., and Jacobi, C.: Non-zonal gravity wave forcing of the Northern Hemisphere winter circulation and effects on middle atmosphere dynamics, EGUsphere, 2025, 1–35, https://doi.org/10.5194/egusphere-2025-3005, 2025a.
- Mitchell, D. M., A. J. Charlton-Perez, and L. J. Gray, 2011: Characterizing the Variability and Extremes of the Stratospheric Polar Vortices Using 2D Moment Analysis. J. Atmos. Sci., 68, 1194–1213, https://doi.org/10.1175/2010JAS3555.1.
- Mitchell, D. M., L. J. Gray, J. Anstey, M. P. Baldwin, and A. J. Charlton-Perez, 2013: The Influence of Stratospheric Vortex Displacements and Splits on Surface Climate. J. Climate, 26, 2668–2682, https://doi.org/10.1175/JCLI-D-12-00030.1.
- Sacha, P., Kuchar, A., Eichinger, R., Pisoft, P., Jacobi, C., & Rieder, H. E. (2021). Diverse dynamical response to orographic gravity wave drag hotspots—a zonal mean perspective. Geophysical Research Letters, 48, e2021GL093305. https://doi.org/10.1029/2021GL093305
- Seviour, W. J. M., D. M. Mitchell, and L. J. Gray (2013), A practical method to identify displaced and split stratospheric polar vortex events, Geophys. Res. Lett., 40, 5268-5273 doi:10.1002/grl.50927.

---

## Author Comment (AC2)

We thank the referee for the insightful comments. Below, we repeat the reviewer's remarks in red italics, and add our respective responses in normal text.

**Reviewer 2**

**General Comments:**

This paper utilizes an unsupervised, hierarchical clustering technique to define geometric clusters representing the behavior of the stratospheric polar vortex as defined by boundary objects and features from a FFT of 10-hPa GPH fields. Thereafter, this work examines the stratospheric polar vortex response to gravity wave forcings in climate model scenarios, with a focus on three hotspot regions: East Asia, the Himalayas, and North America. Their results conclusively show that the hotspot forcings reduce amplitudes of planetary wave size 1 features, with the largest of those changes occurring in the Himalayan hotspot. Additionally, the observed gravity wave forcings hold implications for geometric responses in the stratospheric polar vortex. The results from this paper are compelling and provide interesting contributions to the discussion of stratospheric polar vortex morphology. Outcomes from this paper, specifically regarding the clustering of the vortex, hold applicability for examinations of subseasonal stratospheric-tropospheric teleconnections.

The paper itself is technically dense and difficult to follow at times. It reads like a chapter of a dissertation. The paper would improve greatly with some changes to the wording and extrapolation of specific methodologies to help the content stand as an individual publication. Additionally, the author should explain the underlying motivation for using Fourier descriptors to define the vortex boundary in place of pre-existing, similar methodologies.

We thank the reviewer for their thorough and constructive feedback, as well as for recognizing the contributions of our study. In response to these comments, we have revised the manuscript extensively. These revisions have substantially improved the overall clarity and quality of the manuscript.

I would recommend publication following the completion of these major and minor revisions.

**Major Comments:**

1. There is a lot of "hand waving" done with respect to explaining the specifics of this experiment and its methodology to the point that a lot of backtracking was required to make sense of what was being discussed in Section 3. This is particularly an issue for anything surrounding the model simulations since the author assumes the reader has read Mehrdad et al. 2025a. It is not necessary to include all clarifying information about the specifics of the original experiment in Mehrdad et al. 2025a, but enough detail is needed so that an individual can read this paper on their own and know what is happening. I would suggest that you include more specifics about the methodology

**surrounding model simulations and data in Section 2. Clarified methodological information will improve this section of the paper substantially.**

We appreciate the reviewer's comment and agree that providing clearer context for the model simulations improves readability. The present paper builds directly on Mehrdad et al. (2025a), which provides a full technical description of the experimental design and model configuration. The essential information required to understand the current study—model type, boundary and initial conditions, ensemble size, grid resolution, forcing strength, and the spatial definition of the hotspot regions—is included in Section 2.1 (lines 81-101).

To further clarify the setup while minimizing repetition from the earlier paper, we now provide a concise summary of the distribution of the SSO-induced zonal-wind tendencies by hotspot in Appendix A (lines 486–506 of the revised manuscript). We also added a one-sentence pointer in Section 2.1 directing readers to Appendix A for this summary of the imposed SSO forcing.

Figure A1 in the revised manuscript presents the column-integrated (200–1 hPa) maps of the SSO-induced zonal drag during NDJFM, highlighting where the forcing is applied. Figure A2 shows the corresponding vertical structure of the SSO-induced zonal-mean zonal-wind tendencies for the control and for anomalies in the sensitivity experiments (HI–C, NA–C, EA–C). We believe these additions provide sufficient methodological clarity while avoiding unnecessary redundancy.

2. This paper would benefit from an explanation and motivation for the choice to use Fourier descriptors for diagnosing the vortex boundary as opposed to using pre-existing vortex diagnostic methods, like those from Seviour et al. 2013, or k-means clustering. The FFT method presented here is very compelling, but what is the underlying motivation for using it? Seviour, W. J. M., D. M. Mitchell, and L. J. Gray (2013), A practical method to identify displaced and split stratospheric polar vortex events, Geophys. Res. Lett., 40, 5268-5273 doi:10.1002/grl.50927.

We appreciate this comment and have expanded the motivation in Section 2.3. Our choice of Fourier descriptors is driven by the physics we aim to diagnose: vortex morphology governs where and how planetary and gravity waves interact with the SPV edge. Fourier descriptors offer a compact, geometry-native representation of the boundary in which each coefficient has geometric meaning (roundness, elongation, orientation; finer deformations at higher orders). By retaining only the leading modes, we filter small-scale, transient irregularities and emphasize the large-scale morphology that is dynamically relevant for wave-mean flow coupling, while keeping dimensionality low for robust clustering.

In contrast, threshold diagnostics (e.g., centroid, aspect ratio) provide only a few scalars and cannot distinguish higher-order shape attributes; and gridpoint-wise clustering (e.g., k-means on dynamical fields) measures similarity at fixed locations, which makes it sensitive to modest displacements/rotations and can separate geometrically similar vortices. Our approach therefore retains physical interpretability, increases geometric fidelity, and reduces sensitivity to incidental shifts, which is particularly useful for isolating geometry-conditioned responses to regional GW forcing. (See Section 2.3, lines 112-133).

3. While there is a lot of useful information in the discussion of the sensitivity experiments, sections 3.3-3.5 of this paper suffer from an oversaturation of material. I found it difficult at times to maintain focus on the key takeaways from the figures. This quality of writing is ideal for a dissertation but not necessary for a publication. I would suggest thinning out the text from this area, focusing on the primary results. I may also suggest modifying the figures here to show only the most relevant clusters. For example, the panels in Figures 10-18 for C1, C7, and C10 are generally empty.

We thank the reviewer for this helpful suggestion. In response, we have substantially revised Sections 3.3–3.5 to improve focus and readability. The revised text now emphasizes the primary and consistent signals while foregrounding the key messages from each figure (see revised Sections 3.3–3.5, lines 294-365).

To reduce the density of material, we have also thinned the text and moved the former Figure 8 to Appendix D (now Figure D1; lines 519–532), together with its accompanying description. Furthermore, to enhance readability, we have divided the discussion section into shorter subsections, each focused on a specific aspect while maintaining clear links among them.

Regarding the figures, we fully appreciate the motivation to simplify them by removing panels that convey negligible signals. After careful consideration, we chose to retain the panels for all clusters in each experiment, as they convey meaningful information and are needed for the completeness of the analysis. Showing all clusters allows readers to verify that certain clusters indeed make minimal or inconsistent contributions, which is an important part of the scientific conclusion. Additionally, readers who may be interested in accessing the full set of class contributions—not all of which are mentioned in the text, as we focused the discussion only on the most relevant and consistent clusters—can do so directly from the figures. This approach reduces cognitive load in the narrative while preserving the full evidentiary record.

Overall, these revisions improve clarity and narrative focus while preserving the full evidentiary information conveyed by the figures.

**Minor Comments:**

Line 1-2-I would personally include a reference to what months specifically define the "winter pole" in the Northern Hemisphere.

We thank the reviewer for this suggestion. To address it, we have modified the introduction (line 18 in the revised manuscript).

Line 30 – You may also want to cite this paper as well: Butchart, N. 2022: The Stratosphere: A Review of the Dynamics and Variability. J. of Weather and Climate Dynamics. 3(4), 1237-1272. <a href="https://doi.org/10.5194/wcd-3-1237-2022">https://doi.org/10.5194/wcd-3-1237-2022</a>

We thank the reviewer for this helpful suggestion. The reference to Butchart (2022) has been added in the revised manuscript (line 21).

Line 47 – This could be a technical comment, but I would be inclined to put the initial reference of "compensation mechanisms" in quotes, since it is a reference to terminology.

We thank the reviewer for this suggestion. We have placed "compensation mechanisms" in quotation marks in the revised manuscript (line 47).

**Line 48* – "These" referring to what? The compensation mechanisms?**

We thank the reviewer for this comment. We have clarified the reference by revising the sentence to read "Among these compensation mechanisms is ..." in the revised manuscript (line 48).

**Line 53 – What is meant by "inter-model comparisons of stratospheric dynamics"?* Would it be possible to provide an example?**

We thank the reviewer for this helpful comment. By "inter-model comparisons of stratospheric dynamics," we refer to the large spread that models exhibit in (i) SPV morphological diagnostics (aspect ratio, excess kurtosis, centroid latitude) and (ii) the frequency and ratio of vortex-split versus vortex-displacement SSWs (Sigmond et al., 2023; Kuchar et al., 2024). We have revised the beginning of the paragraph to make these explicit and to improve the logical flow of the paragraph (lines 53–58 in the revised manuscript).

**Line* 59 – "They" referring to what?**

We thank the reviewer for pointing this out. We have revised the sentence by replacing "They" with "These impacts" to clarify the reference and ensure it clearly points to the impacts discussed in the preceding sentence (line 60).

**Line 60-62 - "hotspot-induced changes" ... to GWs? There is a lack of specific clarifying information in this sentence.**

We thank the reviewer for this helpful comment. We have revised the sentence for clarity, changing it from "Previous studies have documented hotspot-induced changes in the zonal-mean context" to "Previous studies have documented the atmospheric response to enhanced GW drag in hotspot regions within a zonal-mean framework" (lines 62-63 in the revised manuscript).

Line 65-72 – As stated in Major Comment 1, some readers will find this paper on its own and may not read Mehrdad et al. 2025a. Silly as it may sound, it is necessary to provide a small amount of context to the reader in the Introduction, one or two sentences, by giving a general overview of Mehrdad et al. 2025a's general goals and key takeaways. The author does a good job of explaining how this specific paper has different motivations, continuing in this paragraph, but does not fully address the previous paper.

We thank the reviewer for this helpful suggestion. We acknowledge that adding this summary improves the paragraph and the overall flow of the introduction. In response, we have inserted a concise overview of Mehrdad et al. (2025a)'s key findings at the beginning of the last paragraph of the Introduction in the revised manuscript (lines 67-71). Additionally, as noted in our response to your earlier comment, Appendix A provides further details on the GW forcing.

Line 65 – Since this is the first mention of UA-ICON, I would reference it in full: "high-top UA-ICON global circulation model experiments."

We thank the reviewer for this helpful suggestion. We have revised the text to read "high-top UA-ICON (ICOsahedral Nonhydrostatic model with Upper-Atmosphere extension)" in the revised manuscript (lines 67-68). We have modify it to high-top UA-ICON (ICOsahedral Nonhydrostatic model with Upper-Atmosphere extension)

Line 67-69 – Something about point (ii), starting with a verb, while points (i) and (iii) start with "we" disrupts the flow of this paragraph and makes it difficult to digest.

We thank the reviewer for this comment. We have revised point (ii) to begin consistently with "we," aligning its structure with points (i) and (iii) (lines 73-75 in the revised manuscript).

Line 71 - I might replace "Arctic" with "Northern Hemisphere" or similar. Or reflect on future nomenclature used in the paper and try to maintain consistency.

We thank the reviewer for this comment. The term "Arctic" has been replaced with "Northern Hemisphere" in the revised manuscript (line 78)

Line 93 — What was the period of record for this winter season data? 30-year simulations are indicated, but what is the start date? Again, this information may be included in the previous paper from Mehrdad et al. 2025a, but it is still important to include clarifying details about the specifics of this experiment as it is its own paper.

We thank the reviewer for this helpful comment. In response, we have added further details about the boundary-condition period, the initialization of the simulations (lines 88-90), and the generation of the ensemble members (lines 97-98). Each experiment consists of six ensemble members, each representing a 30-year simulation (180 model years in total; 175 years for NA see Mehrdad et al. 2025a for more details). After excluding the first simulation year as spin-up, this results in 174 years of analyzed data (169 years for NA). Because the simulations were initialized from and driven by annually repeating present-day climatological conditions, the absolute start date of the integrations is arbitrary and does not affect the results.

**Line 110 – Explain why you retain data northward of 25°N. Similarly, why 18%?**

We thank the reviewer for this comment. We have clarified that data were restricted to latitudes north of 25° N to isolate the Northern Hemisphere extratropical region where the SPV is located (lines 136-137). The choice of the 18 % threshold was determined empirically.

Line 135 – Perhaps be a little more direct/descriptive beyond saying "this feature space" to provide further clarity to the reader.

We thank the reviewer for this suggestion, which enhances clarity. In response, we have revised the sentence on line 163 to more explicitly reference the 24 features.

Line 168-170 — C1 is distinguished for unstable-vortex clusters, in this case meaning more than two or no SPV boundaries. C7-C10 are split vortex events. Would C2-C6 contain a cluster specified for displaced vortex events? What about strong vortex events? You may mention this in detail later, but it is good to give some preliminary indication of what these clusters might contain.

We thank the reviewer for this insightful comment. As the introduced clustering method is hierarchical, and here we only present the top level in the hierarchy, we have retained the content as is. However, to enhance readability and provide a forward reference without preempting the detailed discussion in the results section, we have added a pointer to the relevant part (lines 198-199). This points the reader to the relevant section, where visual content (beyond the dendrogram branching) and in-depth description are presented, maintaining the manuscript's logical flow.

Figure 4 – The cyan lines are difficult to read in this figure. I may suggest using a different colored contour. Additionally, I may separate Panels d and h into their own figures. Included here, the panels are a little small, and their inclusion makes this figure d bit too busy.

Figure 6 – Similar comments to Figure 4. The coloring is difficult to see, and the subplots on this figure are small and difficult to read. Breaking up or structuring this figure differently may make it easier to read.

We thank the reviewers for their constructive feedback. In response, we have revised Figure 4 by removing excess free space, tightening the layout, and enlarging the subplots to improve readability. This adjustment enhances overall clarity without necessitating the separation of panels d and h in Figure 4, as the larger subplots now make the details more visible. We also make the former Figure 6 (now Figure 5 in the revised manuscript) slightly larger in scale in the revised manuscript. Regarding the cyan contours, we evaluated alternative colours but retained the current ones, as they provide the best contrast across the full range of positive and negative anomalies. While they are less distinct over near-zero (white) regions, darker alternatives become indistinguishable against strong anomalies. These changes improve the figures while preserving their comprehensive structure.

**Line 234 – This sentence may not be necessary.**

Thank you for the comment, the sentence is removed in the revised manuscript (line 260).

**Line 262 – What exactly is an "occurrence frequency"/how is it calculated?**

We thank the reviewer for pointing out the need for clarity on the term "occurrence frequency." To address this, we have revised the paragraph (lines 278-279 in the revised manuscript) by adding an explanation of how it is calculated: specifically, as the percentage of days each cluster occurs relative to the total days in the extended winter period.

**Line 375, 377 – "gravity wave" may be replaced with GW.**

We thank the reviewer for this comment. The use of the abbreviation "GW" has been reviewed and applied consistently throughout the revised manuscript

**Line 391 – Which configuration? Be a little more specific.**

We thank the reviewer for this suggestion, which improves the specificity of the description. To address it, we have revised the sentence (lines 392-393 in the revised manuscript) by clarifying "this configuration" to refer explicitly to the equatorward position of the positive anomaly band (north of the forcing region) in the HI experiment.

Line 393 – At this point within the discussion section, I may remind the reader what each of the clusters (C5, C6, etc.) physically represents.

We thank the reviewer for this helpful suggestion, which reinforces the connection between the discussion and the earlier description of the clusters. To address this, we have added a brief reference at the beginning of the paragraph (revised lines 395–396) directing the reader to the section where the cluster characteristics are described in detail. Moreover, we have divided the discussion section into shorter subsections, each focusing on a specific aspect while maintaining clear links among them. These changes improve readability and help the reader more easily follow the key messages and connections across sections.

**Technical Comments:**

In general, I noticed many tense inconsistencies (e.g., present vs. past tense within the same sentence, etc.) Choose one tense to use throughout the paper.

We thank the reviewer for this very helpful comment. We have carefully reviewed the manuscript to ensure consistency in verb tense throughout.

All of the figures would benefit from being increased in resolution. Figure formatting within the text should also be changed to keep the discussion collocated with the image as close as possible.

We thank the reviewer for this comment. We have reviewed all figures in the manuscript and improved their quality where possible by increasing resolution, enlarging scales, and tightening the layout (e.g., Figures 3, 4, B1, and C1). Regarding figure placement, the text discussing each figure is already located as close as possible to the corresponding image in the LaTeX source. We anticipate that any remaining minor spacing issues will be resolved during the journal's typesetting and proofreading process.

**Line 45 – "exert" should be "exerts"**

We thank the reviewer for this observation. Upon review, we confirm that the original verb "exert" is correct, as the compound subject "the location and distribution" is plural and therefore takes a plural verb. Because "location" and "distribution" refer to distinct elements, no change was made to the sentence on line 45.

Line 133 – "features 4 to 23" may be replaced with "features four to twenty-three" for consistency

We thank the reviewer for this suggestion. In the manuscript, we consistently use numerical notation when referring to feature indices. To maintain this consistency, we have chosen to retain the numerical format here (line 160).

**Figure 5 – Increase figure size.**

The figure (now Figure C1 in the revised manuscript) has been enlarged in the revised manuscript.

Section titles 3.3, 3.4, 3.5 – Use the unabbreviated versions of the sensitivity experiments for the titles.

We thank the reviewer for this helpful suggestion. The subsection titles have been revised to use the unabbreviated names of the sensitivity experiments. They now read: Himalayas (HI), Northwest America (NA), and East Asia (EA).

Line 318-320 – "consistent" is used three times in some form here. I may suggest rewording.

We thank the reviewer for this observation. This section has been completely rewritten in the revised manuscript in response to the reviewer's major comment.

```
Line 431, 448 – "zonal mean" should be "zonal-mean" here
```

We thank the reviewer for noting this. The term has been corrected to "zonal-mean" throughout the manuscript for consistency.

```
Line 461 – "help" should be "helps"
```

We thank the reviewer for this observation. Upon review, we confirm that the original verb "help" is correct, as the compound subject "the shape-based clustering and class contribution framework" is plural and therefore takes a plural verb. Because "clustering" and "framework" refer to distinct elements, no change was made to the sentence (line 474).

**References**

Butchart, N.: The stratosphere: A review of the dynamics and variability, Weather and Climate Dynamics, 3, 1237–1272, https://doi.org/10.5194/wcd-3-1237-2022, 2022.

Mehrdad, S., Marjani, S., Handorf, D., and Jacobi, C.: Non-zonal gravity wave forcing of the Northern Hemisphere winter circulation and effects on middle atmosphere dynamics, Weather Clim. Dynam., 6, 1491–1514, https://doi.org/10.5194/wcd-6-1491-2025, 2025a.

Seviour, W. J. M., D. M. Mitchell, and L. J. Gray (2013), A practical method to identify displaced and split stratospheric polar vortex events, *Geophys. Res. Lett.*, 40, 5268-5273 doi:10.1002/grl.50927.

---

## Author Comment (AC3)

We thank the referee for the insightful comments. Below, we repeat the reviewer's remarks in red italics, and add our respective responses in normal text.

**Reviewer 3**

The study addresses the impact of long-term intensified GW forcing on SPV. The authors classified SPV into ten groups based on morphology and analyzed the impact of GW forcing on three major orographic GW hotspots (HI, EA, and NA) separately. Using the UA-ICON global circulation model, the authors conducted multi-ensemble simulations to account for the internal variability of the signal. The paper showed that all hotspots exhibited a consistent decrease in the amplitude of planetary wave number 1 and the corresponding change in circulation. Furthermore, the paper showed that these anomalies are due to changes in the mean SPV structure and frequencies of several clusters, to varying extents depending on the hotspot.

While Mehrdad et al. address highly relevant scientific questions within the scope of ACP, the paper requires minor revisions to be published. The paper introduces a novel methodology for SPV classification and carefully addresses the impact of GW forcing on different clusters. However, the discussions of statistical significance and the mechanism of the signals are lacking and could be improved.

We thank the reviewer for their thoughtful evaluation and constructive feedback. We have carefully addressed the reviewer's concerns, which has improved the overall quality and clarity of the manuscript.

**General comments**

1. Are there any differences in the vertical profiles of GW forcing by the hotspot? Perhaps there are no significant differences, but showing the general picture of how the forcings are applied would be helpful. If differences exist, they could suggest potential reasons for different responses. The example figure in Mehrdad et al. (2025a) helps with understanding, but a more general distribution of the forcing by hotspot would be helpful.

We thank the reviewer for this helpful suggestion. In the revised manuscript, we now provide a concise summary of the zonal-mean vertical distribution of the SSO-induced zonal wind tendencies by hotspot in a new Appendix A (lines 486–506 in the revised manuscript). We also added a one-sentence pointer in Section 2.1 directing readers to Appendix A for this summary of the imposed SSO forcing (lines 98-100).

Figure A1 in the revised manuscript presents the column-averaged (200–1 hPa, stratosphere) maps of the SSO-induced zonal drag during NDJFM, highlighting where the forcing is applied. Figure A2 shows the corresponding vertical structure of the SSO-induced zonal-mean zonal tendencies for the control and for the anomalies in the sensitivity experiments (HI–C, NA–C, EA–C). These diagnostics are adapted from Mehrdad et al. (2025a).

The three hotspots exhibit distinct vertical structures. Over HI and NA, the forcing primarily strengthens the lower-stratospheric (valve-layer) peak, with HI showing a clearer upward

extension and NA remaining more confined in height and accompanied by upper-level westerly anomalies. Over EA, the enhancement is broader and more vertically extended, with less emphasis on a distinct lower-stratospheric maximum. These regional differences reflect variations in GW generation by the SSO scheme and in the background flow.

Because the present paper focuses on the SPV response classified by vortex shape, we keep the treatment of vertical structure brief and refer readers to Mehrdad et al. (2025a) for a full analysis. A dedicated sensitivity study of how the vertical placement of the GW drag influences the SPV would be valuable but lies beyond the scope of the current work.

2. The methodology in the paper is convincing and suggests a novel approach. However, a discussion on why this method was chosen over well-known clustering methods such as EOF, k-clustering, and self-organizing maps would be beneficial. What are the advantages?

Thank you for the positive assessment and for prompting us to clarify the advantages. We have added this discussion to Section 2.3. Traditional EOF/k-means/SOM frameworks operate on gridded fields and therefore optimize gridpoint (or mode) similarity rather than geometric similarity. Consequently, they may assign geometrically similar vortices to different clusters when the structures are shifted or rotated. Our method constructs the feature space from Fourier descriptors of the boundary, yielding a low-dimensional, geometry-aware representation where geometrically similar vortices cluster together, even with modest displacement/orientation differences. This provides (i) richer morphology than a few scalar diagnostics, (ii) greater robustness to small spatial perturbations than gridpoint-wise measures, and (iii) direct physical interpretability of the features themselves. We then apply hierarchical clustering (Ward linkage) in this descriptor space to obtain stable, morphology-based classes, which we use to quantify geometry-conditioned impacts of GW hotspots. (See Section 2.3, lines 112-133)

3. Most of the statistical discussion here relies on the consistency of the signals. However, the significance of these signals (e.g., GPH, zonal wind, and frequency) is questionable. For instance, the authors repeatedly emphasize the strong signal in zonal mean zonal wind, yet the magnitude is less than 1 m/s in most figures. Considering the strong and highly variable winds in the stratosphere, can this signal be significant enough to lead to a meaningful change in circulation?

We thank the reviewer for this insightful comment. We agree that statistical significance tests can provide valuable complementary information. However, in this study, we focus primarily on the consistency of the signals across ensemble members, rather than on the amplitude-based statistical significance. This choice is motivated by the inherently high variability of the stratosphere, where large background fluctuations can obscure physically meaningful but spatially coherent responses.

In our analysis, a signal is considered consistent if at least five out of six ensemble members exhibit anomalies of the same sign as the ensemble mean. This approach highlights robust dynamical responses that persist despite the model's internal variability. Figure 1 (below)

illustrates this for the 10 hPa GPH anomalies (adapted from Figure 3 in the manuscript). The first row shows ensemble-mean anomalies, with dotted regions indicating consistent signals. The second row presents the same anomalies, but shaded only where they are statistically significant based on two-sample t-tests (Student's t for equal variances and Welch's t for unequal variances, with p

Figure 1: Ensemble-mean 10 hPa GPH anomalies for the Himalaya (HI-C, left column), Northwest America (NA-C, middle column), and East Asia (EA-C, right column) forcing experiments relative to the control run. The first row shows the ensemble-mean anomalies with dotted regions indicating areas of consistent response. The second row shows areas where anomalies are statistically significant based on two-sample t-tests (p < 0.05). Green outlines mark the regions of enhanced SSO GW drag.

4. Figure 6 shows that GW forcing seems to explain the zonal wind response over the hotspot and to some extent northward (L230-231). In higher latitudes, however, especially in the HI experiment, EP flux divergence does. What do the authors think led to this difference? L440-441 argues that the delayed adjustment is a potential cause of this inconsistent response, but how can this explain the latitudinal dependence of the balance between GW and resolved waves?

Thank you for highlighting this subtle but important observation. Different forcing components dominate in different regions, including both resolved wave forcing (via EP flux divergence) and parameterized GW drag, encompassing both SSO and non-orographic sources. Changes in the zonal-mean zonal wind induced by any of these components can, in turn, modify the others through feedbacks on the mean flow and residual circulation.

These components are not distributed uniformly in space. In the zonal-mean frame, SSO GW drag typically exhibits a peak in the mid-latitudes near the valve layer, whereas resolved wave momentum deposition (EP flux divergence) tends to dominate at higher latitudes, where stronger wave propagation and convergence occur within the polar stratosphere. This distinction helps explain why GW forcing governs the response near the hotspot and its immediate northward extension, while EP flux divergence becomes dominant at higher latitudes—particularly in the HI experiment. These interactions are examined in detail in Mehrdad et al. (2025a), and the SSO component is now briefly discussed in the new Appendix A, which provides diagnostics of the vertical and latitudinal distribution of SSO GW drag.

Regarding the former Figure 6 (now Figure 5 in the revised manuscript), the anomalies are averaged over the extended winter and across all ensembles, thus reflecting time-integrated (and potentially delayed) responses that combine the effects of these components. As for the discussion at L440–441 (now Section 4.3 lines 445-446), this specifically concerns the C6 WCVC contribution to the HI zonal wind anomalies (Figure 9 in the revised manuscript, panel C6a) and the corresponding EP flux divergence (Figure 10 in the revised manuscript, panel C6a). We initially expected C6, which contributes most strongly to the zonal-mean zonal wind, to also display the largest EP flux divergence signal. However, the class contribution framework captures only instantaneous covariances between fields and cannot represent the delayed adjustments typical of wave—mean flow interactions. We have clarified this limitation in the revised manuscript.

5. Overall, the mechanistic discussion is lacking. Why do we see certain anomalies in WCVC? What causes frequency changes? Why do they respond differently depending on the hotspot? While this may be outside the scope of this study, it would be better to express the authors' opinion and leave it as an open question.

We thank the reviewer for their thoughtful comments. We agree that the Discussion in the preprint was dense, which impeded clear answers to the questions raised. In the revision, we have (i) divided the discussion section into shorter subsections, each focused on a specific aspect while retaining the links among them, and (ii) edited several paragraphs in the discussion to make the central message clearer. The lack of explaining the GW forcing in the paper and purely relying on Mehrdad et al. (2025a) also contributed to obscuring the message in the discussion, which has now been addressed in Appendix A and integrated into the discussion.

Below we address each specific question you raised, with pointers to the relevant parts of the manuscript.

Why do we see certain anomalies in WCVC? What causes frequency changes?

WCVC anomalies arise from forcing-induced internal structural changes within clusters, such as minor shifts in geometry or location that do not reassign days to different clusters, while

frequency changes reflect shifts in cluster prevalence that favor certain morphologies (e.g., low-PW1 configurations). This dual response allows the forcing to both modify existing cluster characteristics and alter their occurrence, contributing to/favouring overall anomalies like PW1 amplitude reduction. See Section 2.4 for the methodological framework and Section 4.2 (paragraphs 1–4) for the application to our results; we have modified paragraphs 3 and 4 to enhance clarity on these mechanisms (lines 394-426).

Why do they respond differently depending on the hotspot?

Responses differ due to the hotspot's spatial position relative to the SPV geometry (with a possible role for the vertical distribution of the forcing), which shapes edge mixing, wave-mean flow interactions, and internal variability. For example, HI's southern position deepens the vortex via edge sharpening, while NA/EA's poleward placement allows deeper core intrusions. See Section 4.2 (paragraph 5, lines 427-434) for geometry-forcing interactions and Section 4.3 (paragraph 1, lines 436-446) for experiment-specific examples, including references to the vertical GW drag distribution in the new Appendix A; we have modified these paragraphs to integrate Appendix A more explicitly and clarify the mechanistic links.

**Minor comments**

**L53: The first sentence of this paragraph does not seem related to the rest of it.**

We thank the reviewer for this comment. We have modified the beginning of the paragraph to improve clarity and ensure that it is better integrated with the rest of the paragraph (lines 53-58 of the revised manuscript)

**L145-146*: *Is rescaling not necessary for the boundary size and time feature?**

We thank the reviewer for this insightful question. Rescaling was not applied to the boundary size or time features because we did not intend to emphasize their role in the clustering process, and their variances are comparable to those of the Fourier descriptors.

**Figure 2. What do different colors mean?**

We thank the reviewer for this observation regarding the dendrogram colors. To improve clarity, we have added a brief explanation to the figure caption in the revised manuscript, noting that the colors represent different cluster branches and their hierarchical relationships.

**L219-221: divergence is reduced -> convergence is reduced?**

We thank the reviewer for this comment. In this context, the phrase "divergence is reduced" is used intentionally (lines 244-246 in the revised manuscript). As described in the sentence, this

refers to a positive EP flux divergence anomaly, meaning that the magnitude of divergence (typically negative, indicating strong resolved wave drag) becomes smaller—i.e., less negative—relative to the control run. This corresponds to weaker wave drag, as noted in the explanatory clause ("which indicates weaker wave drag"). Alternatively, this could also be described as an increase in convergence, but we have retained the original phrasing for consistency with standard usage in the literature.

Figure 7. The GPH differences in C7-C10 are difficult to discern. Therefore, it is difficult to determine if the boundary aligns with the GPH. Is it possible to show the discrepancy more clearly?

We thank the reviewer for this valid point regarding the visibility of GPH differences in clusters C7–C10 of the former Figure 7 (now Figure 6 in the revised manuscript). We acknowledge that the weaker GPH gradients in these split-vortex clusters can make alignment with the boundaries harder to discern. After considering options, such as using cluster-specific colorbars, we decided to retain the current uniform colorbar, as varying scales would complicate inter-cluster comparisons and make the figure busier overall.

L259-262: The first sentence states that C3 has a relatively circular vortex, but a later sentence says that C3 is strongly deformed. This is confusing and needs clarification.

We thank the reviewer for highlighting this. To resolve this, we have revised the paragraph (now located in Appendix D, lines 530-532 in the revised manuscript) to clearly distinguish C6 as circular and to emphasize that C3's deformation is mainly associated with PW1.

L349: How is SSW linked to C4 and C8? There is no earlier discussion about this. It would be better to either remove the statement or show the relationship in the supplementary material.

We thank the reviewer for this helpful comment. The SSW events were introduced in the Introduction (lines 30-33) and now explicitly highlighted in Section 3.2 of the revised manuscript (lines 266-267, 287-288) to clarify that they refer to the shape-based identification of displaced and split vortex configurations.

L383-392: First, it is argued that positive GPH anomalies north of the hotspot lead to the weakening and mixing of the SPV edge. However, for the HI experiment, a similar signal leads to the sharpening of the SPV. This needs to be clarified, as a similar signal led to a different response, and the two arguments conflict with each other.

We thank the reviewer for highlighting this point. To clarify, we have revised the paragraph (lines 392-393 in the revised manuscript) to emphasize that the apparent difference arises from the equatorward position of the positive anomaly band in the HI experiment—located south of

the vortex edge over northern Eurasia—which leads to a sharpening rather than a weakening of the SPV. The pattern of a positive anomaly in a higher-latitude band relative to the forcing and a negative anomaly within the forcing region is consistent across all experiments; however, the relative position of these anomalies with respect to the SPV determines the nature of the vortex response (see also lines 427-434 of the revised manuscript).

**L405-407: Same here. The SSW linkage to C4 and C8 must be shown explicitly first.**

We thank the reviewer for this comment. As noted in our response to the previous comment, we have added a brief explanation in Section 3.2 of the revised manuscript.

**L426-428: Could you clarify what it means to "follow mean circulation"?**

We thank the reviewer for this comment. To address this, we have added a concise parenthetical explanation in the revised sentence (lines 430-431), clarifying that "following the mean circulation" refers to the geostrophic wind direction along the mean GPH isolines, as noted in the preceding sentence (lines 428-430).

**L428-429: Again, the strengthening of the SPV edge conflicts with the sentence before. Please clarify this.**

We thank the reviewer for this comment. As noted in our previous response, the effect of the forcing on the SPV depends on its latitudinal position relative to the vortex edge. Although a positive GPH anomaly in a higher-latitude band relative to the forcing is consistently found across experiments, its influence on the SPV differs depending on whether the anomaly lies equatorward or poleward of the vortex edge.

**L442: C9 -> C8**

Th We thank the reviewer for this observation. This has been corrected in the revised manuscript (line 447).

**L444-446*: Positive anomalies seem to exist at different longitudes.**

We thank the reviewer for this observation. To address the comment, we have revised the sentence (lines 449-450 in the revised manuscript) to clarify that these anomalies occur in a higher-latitude band north of the forced region.

**References**

Mehrdad, S., Marjani, S., Handorf, D., and Jacobi, C.: Non-zonal gravity wave forcing of the Northern Hemisphere winter circulation and effects on middle atmosphere dynamics, Weather Clim. Dynam., 6, 1491–1514, https://doi.org/10.5194/wcd-6-1491-2025, 2025a.